# Quantum Kernelized Bandits

**Yasunari Hikima**[*1]     **Kazunori Murao**[*1]     **Sho Takemori**[*1]     **Yuhei Umeda**[*1]

[1]AI Laboratory, Fujitsu Limited, Kawasaki, Japan

## Abstract

We consider the quantum kernelized bandit problem, where the player observes information of rewards through quantum circuits termed the quantum reward oracle, and the mean reward function belongs to a reproducing kernel Hilbert space (RKHS). We propose a UCB-type algorithm that utilizes the quantum Monte Carlo (QMC) method and provide regret bounds in terms of the decay rate of eigenvalues of the Mercer operator of the kernel. Our algorithm achieves $\widetilde{O}\left(T^{\frac{3}{1+\beta_p}}\log\left(\frac{1}{\delta}\right)\right)$ and $\widetilde{O}\left(\log^{3(1+\beta_e^{-1})/2}(T)\log\left(\frac{1}{\delta}\right)\right)$ cumulative regret bounds with probability at least $1-\delta$ if the kernel has a $\beta_p$-polynomial eigendecay and $\beta_e$-exponential eigendecay, respectively. In particular, in the case of the exponential eigendecay, our regret bounds exponentially improve that of classical algorithms. Moreover, our results indicate that our regret bound is better than the lower bound in the classical kernelized bandit problem if the rate of decay is sufficiently fast.

## 1 INTRODUCTION

Quantum machine learning is an emerging research field that attempts to enhance machine learning methods with quantum technology [Biamonte et al., 2017, Dunjko and Briegel, 2018, Schuld and Petruccione, 2018, Gyongyosi and Imre, 2019]. The primary objective of quantum machine learning is to accelerate and improve the performance of classical machine learning algorithms using quantum computing paradigms and techniques. For instance, Grover's algorithm [Grover, 1996], which is a well-known quantum algorithm for solving the problem of finding a unique item

---

[*]Alphabetical order.

from an unstructured database of $N$ items, succeeded in reducing the time complexity to $O(\sqrt{N})$, while the classical method has a time complexity $O(N)$.

The study of quantum algorithms for the bandit problems has also attracted attention in the field of machine learning, and there is much interest in the quantum acceleration of classical algorithms that have been studied so far ([Gyongyosi and Imre, 2019, Biamonte et al., 2017]). Many quantum algorithms for the classical bandit problems have been studied for various settings including best-arm identification ([Casalé et al., 2020, Wang et al., 2021]), exploration-exploitation with stochastic environments ([Wan et al., 2023]), and adversarial environments ([Cho et al., 2023]).

Following Wan et al. [2023], this paper focuses on a sequential decision-making problem called the *quantum bandit problem*. For a given fixed set of actions $\mathcal{X}$, for each round $t = 1, 2, \dots, T$, the player chooses an action $x_t \in \mathcal{X}$. The objective of the player is to maximize the cumulative reward $\sum_{t=1}^{T}\mu(x_t)$, and the performance is measured in terms of the cumulative regret over $T$ rounds, which is defined as $R(T) = \sum_{t=1}^{T}\left(\mu(x^\star) - \mu(x_t)\right)$, where $\mu : \mathcal{X} \to [0, 1]$ is the mean reward function, and $x^\star \in \operatorname{argmax}_{x \in \mathcal{X}}\mu(x)$ is the best action in hindsight. During the game, the player has a chance to access the unitary operator (quantum circuit) $\mathcal{O}_x$ or its adjoint $\mathcal{O}_x^\dagger$, referred to as *quantum reward oracle*, that encodes the reward distribution associated with the action. Invoking the quantum reward oracle and performing a measurement, the player can obtain the information about the reward but the number of query calls is limited up to $T$. One can apply any classical bandit algorithm to this problem setting, however, since the player can utilize quantum algorithms, an algorithm designated for this problem setting could perform much better. We review the details of the notion in Sec. 3.2.

Wan et al. [2023] studied the case where the reward function $\mu(x)$ is linear with respect to an action $x$. By adapting the quantum Monte Carlo [Montanaro, 2015] (QMC) method, they proposed an algorithm called QLinUCB that attains

Table 1: Comparison of the cumulative regret upper bounds with the classical and quantum algorithm, where $\beta_p > 1, \beta_e > 0$ are constants, $\delta \in (0,1)$ is the probability level, and $T > 0$ is the total number of rounds, $\alpha > 0$ is any positive number.

| Reference | Kernel | Setting | Regret bound |
|---|---|---|---|
| Vakili et al. [2021] | $\beta_p$ polynomial eigendecay | Classical | $O\left(T^{\frac{\beta_p+1}{2\beta_p}} \log^{1-\frac{1}{2\beta_p}}(T)\right)$ |
| | $\beta_e$ exponential eigendecay | Classical | $O\left(T^{\frac{1}{2}} \log^{1+\frac{1}{2\beta_e}}(T)\right)$ |
| Wan et al. [2023] | $d$-dimensional linear | Quantum | $O\left(d^2 \log^{5/2} \log(T)\right)$ |
| This paper (Theorem 5.4) | $\beta_p$ polynomial eigendecay | Quantum | $\widetilde{O}\left(T^{\frac{3}{1+\beta_p}} \log\left(\frac{1}{\delta}\right)\right)$ |
| | $\beta_e$ exponential eigendecay | Quantum | $\widetilde{O}\left(\log^{3(1+\beta_e^{-1})/2}(T) \log\left(\frac{1}{\delta}\right)\right)$ |

$O(\text{poly}(\log T))$ regret bound. We extend the work of Wan et al. [2023] to the case when the mean reward function belongs to a reproducing kernel Hilbert space (RKHS) associated to a kernel $k$. In the classical setting, the kernelized bandit problem is also known as Bayesian optimization. In the literature on the classical bandit problem, researchers often consider an extension of linear bandits to the kernelized case. This is possible since a confidence interval for the mean reward estimation is known in the kernelized case [Srinivas et al., 2010, Chowdhury and Gopalan, 2017], and can be used to derive regret bound in the same manner as in the linear case. However, in the quantum kernelized bandit problem, neither confidence intervals nor theoretical properties of algorithms designated for this problem setting have not been well studied.

**Contributions.** This paper extends the quantum linear bandit problem [Wan et al., 2023] to the kernelized case. In this study, we consider the case where the rate of decay of eigenvalues of the Mercer operator is polynomially or exponentially fast and provides an upper bound of the cumulative regret. For instance, Matérn-$\nu$ kernels have a $1 + 2\nu/d$ polynomial eigendecay, and squared exponential (SE) kernels have a $1/d$ exponential eigendecay, if $\mathcal{X} \subset \mathbb{R}^d$. We show that the proposed algorithm achieves $\widetilde{O}\left(T^{\frac{3}{1+\beta_p}} \log\left(\frac{1}{\delta}\right)\right)$ regret bound if the kernel $k$ has a $\beta_p$ polynomial eigendecay, and $\widetilde{O}\left(\log^{3(1+\beta_e^{-1})/2}(T) \log\left(\frac{1}{\delta}\right)\right)$ if the kernel $k$ has a $\beta_e$ polynomial eigendecay. This result indicates that the proposed method exponentially improves compared with that of classical algorithms [Valko et al., 2013, Vakili et al., 2021] under the condition of the exponential eigendecay. We summarized the relevant study in Table 1. We shall defer all the omitted proofs to Appendix.

**Comparison to Quantum Bayesian Optimization [Dai et al., 2023]** Recently, Dai et al. [2023] extended the linear reward model [Wan et al., 2023] to the kernelized case, which is the same problem setting as this paper. Compared to [Dai et al., 2023], this paper has the following advantages. **(i) Theoretical analysis without the unbiasedness assumption of the QMC estimator**. Dai et al. [2023] pro-

vided regret upper bounds (e.g., $\widetilde{O}(\log^{3(d+1)/2}(T))$ in the case of squared exponential kernels), however, their proof implicitly assumes that the quantum Monte Carlo method [Montanaro, 2015] is an unbiased estimator. As we will detail in Sec. 6.1.1, this assumption is unlikely to hold. Our regret bounds do not rely on the unbiasedness assumption. Thus we provide more mathematically rigorous analysis compared to [Dai et al., 2023]. **(ii) Improved regret bounds**. Even under the unbiasedness assumption of the QMC estimator, our regret bound $\widetilde{O}\left(T^{\frac{3}{1+\beta_p}} \log\left(\frac{1}{\delta}\right)\right)$ is better than that $\widetilde{O}\left(T^{\frac{3}{\beta_p}} \log\left(\frac{1}{\delta}\right)\right)$ of Q-GP-UCB [Dai et al., 2023], in the case of the $\beta_p$-polynomial eigendecay. **(iii) A novel tradeoff parameter** $\eta$. Both our algorithm (Algorithm 1) and Q-GP-UCB [Dai et al., 2023] extend QLinUCB [Wan et al., 2023] to the kernelized case and these algorithms divide the time interval into several stages. There is a tradeoff between the total number of stages and regret incurred in each stage. We not only extend QLinUCB to the kernelized case, but also introduce a novel tradeoff parameter $\eta$, which is a key feature that leads to the aforementioned improved regret bounds. **(iv) A novel proof technique for bounding the (weighted) information gain**. Both this paper and Dai et al. [2023] provide an upper bound of the "weighted information gain" $\gamma^{\text{QMC}}$ (see (3) for definition and Corollary 5.6), which is an analogue of [Vakili et al., 2021, Theorem 3]. While Dai et al. [2023] almost repeated the proof of [Vakili et al., 2021, Theorem 3], we provide a more generalized result (Proposition 5.5) including these results. In particular, our proof provides a simple alternative proof of [Vakili et al., 2021, Theorem 3].

## 2 PRELIMINARIES OF QUANTUM COMPUTATION

Following [Nielsen and Chuang, 2010], we briefly review the basic notion of quantum computation. Then, we introduce quantum Monte Carlo method [Montanaro, 2015], which provides quadratic speedup compared to the classical mean estimator.

## 2.1 BASICS OF QUANTUM COMPUTATION

Let $\mathcal{V}$ be a finite dimensional complex vector space with an hermitian inner product $\langle \cdot, \cdot \rangle$, i.e., a finite dimensional Hilbert space over the complex field $\mathbb{C}$. We call a such space a state space. A state vector (or a quantum state) is an element of the state space $\mathcal{V}$ with a unit norm. Using the bra-ket notation, we denote a state vector in $\mathcal{V}$ by $|x\rangle$. For a state vector $|x\rangle$, we denote by the corresponding vector $|x\rangle^{\dagger}$ in the dual space of $\mathcal{V}$ (which can be identified with $\mathcal{V}$ itself) by $\langle x|$. For a hermitian operator $H : \mathcal{V} \to \mathcal{V}$, a state vector $|x\rangle$, and a vector $\langle y|$ in the dual space, we denote the inner product of $\langle y|$ and $H |x\rangle$ by $\langle y|H|x\rangle$. For two state spaces $\mathcal{V}_1, \mathcal{V}_2$, we can naturally regard the tensor product $\mathcal{V}_1 \otimes \mathcal{V}_2$ as a state space and we simply denote the tensor product $|x_1\rangle \otimes |x_2\rangle$ by $|x_1\rangle |x_2\rangle$, where $|x_1\rangle \in \mathcal{V}_1$ and $|x_2\rangle \in \mathcal{V}_2$.

A quantum computer (or quantum mechanics) does not provide us complete information of a state vector. Instead, one can observe partial information of a state vector by performing a *measurement*. More formally, quantum measurements are described by a collection of linear operators $\{M_{\sigma} : \mathcal{V} \to \mathcal{V}\}_{\sigma \in \Lambda}$ called the measurement operators, where $\Lambda$ is the space of outcomes (we will provide an example below). We assume that the collection of measurement operators satisfies the following completeness equation: $\sum_{\sigma \in \Lambda} M_{\sigma}^{\dagger} M_{\sigma} = I_{\mathcal{V}}$. Here, $M_{\sigma}^{\dagger}$ denotes the adjoint of the operator $M_{\sigma}$ and $I_{\mathcal{V}}$ denotes the identity operator of $\mathcal{V}$. Assuming that $|x\rangle$ represents the state vector immediately before the measurement, the probability $p(\sigma)$ that an outcome $\sigma$ occurs is given as follows [Nielsen and Chuang, 2010, Chapter 2.2.3]: $p(\sigma) = \langle x|M_{\sigma}^{\dagger} M_{\sigma}|x\rangle$, We note that the completeness equation assures the equality $\sum_{\sigma \in \Lambda} p(\sigma) = 1$. Moreover, the state vector after the measurement is given as $\frac{M_{\sigma}|x\rangle}{\sqrt{\langle x|M_{\sigma}^{\dagger} M_{\sigma}|x\rangle}}$.

Given an initial state vector $|x_0\rangle$, a *quantum algorithm* sends the vector $|x_0\rangle$ to $U_k U_{k-1} \cdots U_1 |x_0\rangle$, and performs a measurement to obtain an outcome, where $U_1, \ldots, U_k$ are unitary operators.

**Example 2.1** (Measurements of a qubit in the computational basis). We provide a simple example of a state space and measurement operators. Let $\mathbb{C}^2$ be the $\mathbb{C}$-Hilbert space with the canonical inner product and we denote by $|0\rangle$ and $|1\rangle$ the canonical basis of $\mathbb{C}^2$. For a positive integer $n \in \mathbb{Z}_{\geq 1}$, we define a state space $\mathcal{V}_n$ by $(\mathbb{C}^2)^{\otimes n}$ and define the outcome space $\Lambda_n$ the set of binary sequences of length $n$, i.e., $\Lambda_n = \{a_1 a_2 \ldots a_n : a_i \in \{0,1\}, i = 1, \ldots, n\}$. Then, a set $\mathcal{B}_n = \{|\sigma\rangle \in \mathcal{V}_n : \sigma \in \Lambda_n\}$ forms an orthonormal basis of $\mathcal{V}_n$, where for $\sigma = a_1 a_2 \cdots a_n \in \Lambda_n$, we define $|\sigma\rangle$ by $|a_1\rangle |a_2\rangle \cdots |a_n\rangle$. The set of state vectors is given as $\{\sum_{\sigma \in \Lambda_n} \alpha_{\sigma} |\sigma\rangle : \sum_{\sigma \in \Lambda_n} |\alpha_{\sigma}|^2 = 1\}$. For $\sigma = \Lambda_n$, we define a linear operator $M_{\sigma} : \mathcal{V}_n \to \mathcal{V}_n$ by $M_{\sigma} = |\sigma\rangle \langle \sigma|$, i.e., $M_{\sigma}(|\sigma'\rangle) = \delta_{\sigma\sigma'}$ for $\sigma' \in \Lambda_n$, where $\delta$ denotes the Kronecker $\delta$. Then, the collection of operators $\{M_{\sigma} : \sigma \in \Lambda_n\}$ satisfies the completeness equation. With this measurement operators, for a state vector $\sum_{\sigma \in \Lambda_n} \alpha_{\sigma} |\sigma\rangle$, an outcome $\sigma$ is observed with probability $|\alpha_{\sigma}|^2$ and after the measurement the state vector collapses into another state.

## 2.2 QUANTUM MONTE CARLO

Following Rebentrost et al. [2018], we introduce quantum Monte Carlo methods [Montanaro, 2015]. Let $n$ be a positive integer, and let $\mathcal{V}_n$ and $\Lambda_n$ be the state space $(\mathbb{C}^2)^{\otimes n}$ and the set of binary sequences of length $n$, respectively as in Example 2.1. Let $v : \Lambda_n \to [0, 1]$ be a function. Let $y$ be a random variable taking values in $[0, 1]$ assume that the probability $P(y = v(\sigma))$ is given as $p(\sigma)$ for each $\sigma \in \Lambda_n$. Let us suppose that we are interested in an estimator of the expectation $\mathbb{E}[y] = \sum_{\sigma \in \Lambda_n} p(\sigma) v(\sigma)$.

We assume that an implementation of the random variable $y$ is given in a quantum computer. More formally, we assume that there exists a unitary operator $\mathcal{O}(y)$ acting on $\mathcal{V}_{n+1}$ and $\mathcal{O}(y) |0^{n+1}\rangle$ is given as

$$\sum_{\sigma \in \Lambda_n} \sqrt{p(\sigma)} |\sigma\rangle \left( \sqrt{1 - v(\sigma)} |0\rangle + \sqrt{v(\sigma)} |1\rangle \right) \quad (1)$$

Here $|0^{n+1}\rangle = |0\rangle \cdots |0\rangle \in \mathcal{V}_{n+1}$. By performing a measurement of $\mathcal{O}(y) |0^{n+1}\rangle$ with the computational basis, we observe an outcome $\sigma 1$ with probability $v(\sigma) p(\sigma)$ and outcome $\sigma 0$ with probability $(1 - v(\sigma)) p(\sigma)$. Therefore, if we define a random variable $B_y$ by $B_y = 1$ if an outcome is $\sigma 1$ for some $\sigma \in \Lambda_n$ (i.e., the last qubit is 1) and $B_y = 0$ otherwise, then we have $\mathbb{E}[B_y] = \mathbb{E}[y] = \sum_{\sigma \in \Lambda_n} p(\sigma) v(\sigma)$. Thus, by repeating this procedure, and taking an empirical mean of observations of $B_y$, one can obtain an estimator of $\mathbb{E}[y]$. However, to make the estimation error smaller than $\epsilon$, we have to call the unitary operator $O(1/\epsilon^2)$ times. Quantum Monte Carlo methods [Montanaro, 2015] provide quantum speedup compared to the classical estimator.

**Lemma 2.2** (Quantum Monte Carlo Method, Montanaro [2015]). *Let $y$ be a random variable that takes values in $[0, 1]$ and $\mathcal{O}(y)$ the unitary operator as in (1). Let $\epsilon > 0$ and $\delta \in (0, 1)$. Then, there exists a constant $C > 1$ and a quantum algorithm $\text{QMC}(\mathcal{O}(y), \epsilon, \delta)$ that outputs a mean estimate $\hat{y}$ satisfying the following conditions.*

1. *$P(|\hat{y} - \mathbb{E}[y]| \geq \epsilon) \leq \delta$.*
2. *The quantum algorithm $\text{QMC}(\mathcal{O}(y), \epsilon, \delta)$ queries the unitary operator $\mathcal{O}(y)$ or its adjoint $\mathcal{O}(y)^{\dagger}$ at most $\frac{C}{\epsilon} \log(1/\delta)$ times.*

More precisely, it is assumed that quantum Monte Carlo methods can operate state vectors in a composite system $\mathcal{V}_n \otimes \mathcal{V}_{n'}$ rather than $\mathcal{V}_n$, where $n' \geq 1$ is a fixed integer, i.e., it is assumed that additional $n'$ qubits are available. We note that one can operate the unitary operator $\mathcal{O}(y)$ on $\mathcal{V}_n \otimes \mathcal{V}_{n'}$ by $\mathcal{O}(y) \otimes 1_{\mathcal{V}_{n'}}$. We refer to [Rebentrost et al., 2018, Montanaro, 2015] for further details.

## 3 PROBLEM FORMULATION

In this section, we introduce a concept of quantum reward oracle and provide a formal descrition of the quantum bandit problem. We also introduce the mean reward function and the reproducing kernel Hilbert space (RKHS) associated with the kernel. Finally, we briefly review Mercer's theorem, which is a key tool for our theoretical analysis.

### 3.1 QUANTUM REWARD ORACLE AND QUANTUM BANDITS

Following [Wan et al., 2023], we introduce a notion of quantum reward oracle. Let $\mathcal{X}$ be a set of actions. We let $n \in \mathbb{Z}_{\geq 1}$ and consider the space $\mathcal{V}_n$ of $n$ qubits and the set $\Lambda_n$ of binary sequences. For each action $x \in \mathcal{X}$, let $y_x$ be a random variable taking values in $[0, 1]$ such that $P(y_x = v_x(\sigma)) = p_x(\sigma)$ for $\sigma \in \Lambda_n$, where $v_x : \Lambda_n \to \mathbb{R}$ is a function and $(p_x(\sigma))_{\sigma \in \Lambda_n}$ is a probability measure on $\Lambda_n$. As in Sec. 2.2, we assume that a unitary operator $\mathcal{O}_x$ on $\mathcal{V}_{n+1}$ is given and $\mathcal{O}_x |0^{n+1}\rangle$ is given as :

$$\sum_{\sigma \in \Lambda_n} \sqrt{p_x(\sigma)} |\sigma\rangle \left( \sqrt{1 - v_x(\sigma)} |0\rangle + \sqrt{v_x(\sigma)} |1\rangle \right).$$

We assume that the expected reward associated to an action $x \in \mathcal{X}$ is given as $\mathbb{E}[y_x]$ and we define the mean reward function $\mu : \mathcal{X} \to [0, 1]$ as $\mu(x) = \mathbb{E}[y_x]$. In Sec. 3.2, we shall detail assumptions on the mean reward function $\mu$. Following [Wan et al., 2023], we call the operator $\mathcal{O}_x$ or its adjoint $\mathcal{O}_x^\dagger$ a quantum reward oracle.

In this paper, we consider the following sequential decision making problem. For each round $t = 1, \ldots, T$, a player selects an action $x_t \in \mathcal{X}$ and incurs an instantaneous regret $\mu(x^*) - \mu(x_t)$, where $x^* = \operatorname{argmax}_{x \in \mathcal{X}} \mu(x)$. During the process, the player can invoke any unitary operators and perform a measurement, however, we assume that the number of calls of quantum reward oracles $\mathcal{O}_x, \mathcal{O}_x^\dagger$ is limited up to $T$. The objective of the player is to minimize the cumulative regret defined as $R(T) = \sum_{t=1}^{T} (\mu(x^*) - \mu(x_t))$.

We note that we can apply any classical bandit algorithm for regret minimization to our problem setting. More precisely, for each round $t$ and a selected action $x_t$, by invoking the quantum reward oracle $\mathcal{O}_{x_t}$ and performing a measurement, we observe a random reward $\mu(x_t) + \varepsilon_t \in [0, 1]$ with the expectation $\mu(x_t)$. Therefore, based on observed rewards $\mu(x_1) + \varepsilon_1, \ldots, \mu(x_{t-1}) + \varepsilon_{t-1}$, any classical bandit algorithm can select an action $x_t$ to minimize the cumulative regret. However, in our problem setting, the player can perform quantum computation with a limited number of oracle queries. Therefore, an optimal algorithm for quantum bandits potentially could perform much better than classical bandit algorithms in terms of cumulative regret.

### 3.2 MEAN REWARD FUNCTION AND RKHS

We make the following assumption on the mean reward function $\mu : \mathcal{X} \to [0, 1]$ [1]. Let $k : \mathcal{X} \times \mathcal{X} \to \mathbb{R}$ be a semi-positive definite kernel. We denote by $\mathcal{H}_k$ the RKHS corresponding to the kernel $k$, i.e., $\mathcal{H}_k$ is the subspace of real valued functions on $\mathcal{X}$ satisfying the following three conditions. (i) $\mathcal{H}_k$ is a real Hilbert space. (ii) For any $x' \in \mathcal{X}$, the feature vector $\phi(x')$ belongs to $\mathcal{H}_k$, where $\phi(x')$ is a function on $\mathcal{X}$ defined as $x \mapsto k(x, x')$. (iii) For any $f \in \mathcal{H}_k$ and $x \in \mathcal{X}$, we have $\langle f, \phi(x) \rangle = f(x)$. The last property is called the reproducing property. We call the map $\mathcal{X} \ni x \mapsto \phi(x) \in \mathcal{H}_k$ the feature map of the RKHS. For $f, g \in \mathcal{H}_k$, we denote the inner product $\langle f, g \rangle$ by $f^\top g$. We assume that the mean reward function $\mu$ belongs to $\mathcal{H}_k$, i.e., there exists $\theta^* \in \mathcal{H}_k$ such that $\mu(x) = \phi(x)^\top \theta^*$.

Examples of kernels defined on $\mathbb{R}^d \times \mathbb{R}^d$ includes the squared exponential (SE) kernels, Matérn-$\nu$ kernels, and rational quadratic (RQ) kernels. Let $l > 0$ be a length-scale parameter. A SE kernel $k_{\text{SE}}$ is defined by $k_{\text{SE}}(x, y) = \exp\left(-\|x - y\|_2^2/l\right)$, where $x, y \in \mathbb{R}^d$. A Matérn kernel $k_{\text{Matérn}}$ is defined on $\mathbb{R}^d \times \mathbb{R}^d$ by $k_{\text{Matérn}}(x, y) = \frac{2^{1-\nu}}{\Gamma(\nu)} (a\sqrt{2\nu})^\nu K_\nu(a\sqrt{2\nu})$ for $x, y \in \mathbb{R}^d$, where $\nu > 0$ is a smoothness parameter, $a = \|x - y\|_2/l$, and $K_\nu$ is the modified Bessel function of the second kind. A RQ kernel $k_{\text{RQ}}$ is defined by $k_{\text{RQ}}(x, y) = \left(1 + \|x - y\|_2^2/(2\nu l^2)\right)^{-\nu}$ for $x, y \in \mathbb{R}^d$, where $\nu > 0$ is a parameter.

### 3.3 MERCER'S THEOREM

As we stated in the introduction, our regret bounds involve the decay rate of the eigenvalues of the Mercer operator. Here, following [Steinwart and Christmann, 2008, Chapter 4.5], we briefly review the theoretical properties of the Mercer operator. Let $\mathcal{X}$ be a measurable space and $\nu$ be a finite measure on $\mathcal{X}$, and $k : \mathcal{X} \times \mathcal{X} \to \mathbb{R}$ be a measurable kernel. We denote by $L_2(\nu)$ the space of square-integrable functions on $\mathcal{X}$ with respect to the measure $\nu$. We define an integral operator $\mathcal{T}_k : L_2(\nu) \to L_2(\nu)$ called the Mercer operator by $f \mapsto \int_{\mathcal{X}} k(\cdot, x) f(x) d\nu(x)$. Since $\mathcal{T}_k$ is compact, positive, and self-adjoint, by the spectral theorem, there exists an orthonormal basis $\{\psi_i\}_{i \in I}$ of $L_2(\nu)$ such that for any $f \in L_2(\nu)$, $\mathcal{T}_k$ has the following expansion $\mathcal{T}_k f = \sum_{i \in I} \lambda_i \langle \psi_i, f \rangle_{L_2(\nu)} \psi_i$. Here, $\{\lambda_i\}_{i \in I}$ is a set of non-zero eigenvalues of $\mathcal{T}_k$ with $\lambda_1 \geq \lambda_2 \geq \cdots > 0$. We refer to [Steinwart and Christmann, 2008, Theorem 4.49 and 4.51] for the following form of Mercer's theorem.

**Theorem 3.1** (Mercer's Theorem). *Let $\{\psi_i\}_{i \in I}$ and $\{\lambda_i\}_{i \in I}$ be defined as above. Assume that $\mathcal{X}$ is a compact metric space, $k : \mathcal{X} \times \mathcal{X} \to \mathbb{R}$ is a continuous kernel, and $\nu$ is a finite Borel measure with $\operatorname{supp} \nu = \mathcal{X}$. Then, we have*

---

[1] We refer to a remark after Assumption 5.1 for validity of this assumption.

*the following expansion:*

$$k(x, x') = \sum_{i \in I} \lambda_i \psi_i(x) \psi_i(x'), \quad x, x' \in \mathcal{X}.$$

*Here, the convergence is absolute and uniform. Moreover, $\{\lambda_i^{1/2} \psi_i\}_{i \in I}$ forms an orthonormal basis of $\mathcal{H}_k$.*

To discuss the theoretical property of our algorithm (Algorithm 1 in Sec. 4), we introduce the following formal characteristic of eigendecay as defined in Chatterji et al. [2019, Definition 11] and Vakili et al. [2021, Definition 1]:

**Definition 3.2** (Eigen-decay). Let $\{\lambda_i\}_{i \in I}$ be the eigenvalues of the Mercer operator with $\lambda_1 \geq \lambda_2 \geq \cdots > 0$ and $I \subseteq \mathbb{Z}_{\geq 1}$ as in Theorem 3.1.

1. Let $C_p > 0$ and $\beta_p > 1$ be constants. We say a kernel $k$ has a $(C_p, \beta_p)$ polynomial eigendecay, if for all $n \in I$, we have $\lambda_n \leq C_p n^{-\beta_p}$.

2. Let $C_{e,1} C_{e,2} > 0$ and $\beta_e > 0$ be constants. We say a kernel $k$ has a $(C_{e,1}, C_{e,2}, \beta_e)$ exponential eigendecay, if for all $n \in I$, we have $\lambda_n \leq C_{e,1} \exp(-C_{e,2} n^{\beta_e})$.

If we ignore constants $C_p, C_{e,1}, C_{e,2}$, then we simply say $k$ has a $\beta_p$ polynomial eigendecay or $\beta_e$ exponential eigendecay.

We provide examples of eigendecay of kernels in the case when $\mathcal{X}$ is a compact subset of $\mathbb{R}^d$. It is known that a Matérn kernel with a smoothness parameter $\nu > 0$ has $(2\nu + d)/d$ polynomial eigendecay [Santin and Schaback, 2016, Theorem 15]. If $k$ is an SE or RQ kernel, then $k$ has $1/d$ exponential eigendecay. The latter statement follows from [Santin and Schaback, 2016, Theorem 15] and [Wendland, 2004, Theorem 11.22].

# 4 A UCB-TYPE ALGORITHM AND CONFIDENCE INTERVAL

In this section, we present a UCB-type algorithm termed QMCKernelUCB as illustrated in Algorithm 1 for the quantum bandit problem with a kernelized reward function. We also introduce a confidence interval of our reward estimator (Proposition 4.2).

## 4.1 PROPOSED METHOD

To leverage the quadratic speedup of the QMC method (Lemma 2.2), we divide the time interval into several stages, which is similar to the doubling trick (c.f., [Lattimore and Szepesvári, 2020, Chapter 6]). For each stage $s = 1, 2, \ldots$, QMCKernelUCB plays an action $x_s \in \mathcal{X}$, and calls the QMC method $\mathrm{QMC}(\mathcal{O}_{x_s}, \eta \epsilon_s, \delta/M)$ with the error tolerance $\eta \epsilon_s$, and observes of an output $y_s$ of the QMC method,

where $x_s$ is an "optimistic estimation" of the best action $x^\star$, and $M, \eta$ are parameters of QMCKernelUCB. We explain how to select the error $\epsilon_s$, the action $x_s$ below (and explain the parameter $\eta$ in Section 4.3). Since the QMC method calls the quantum reward oracle $\mathcal{O}_{x_s}, \mathcal{O}_{x_s}^\dagger$ for $\frac{C}{\eta \epsilon_s} \log(M/\delta)$ times, $x_s$ is an optimistic estimation of the best action at stage $s$, the algorithm plays the same action $x_s$ for successive $\frac{C}{\eta \epsilon_s} \log(M/\delta)$ rounds in stage $s$. Due to the problem setting, it terminates if it consumes $T$ oracle queries.

Because an output $y_s$ with a small estimation error $\epsilon_s$ is more informative than those with larger errors, we consider the following weighted least estimation of the ground truth vector $\theta^* \in \mathcal{H}_k$ with weights $1/\epsilon_i^2$:

$$\hat{\theta}_s \in \operatorname*{argmin}_{\theta \in \mathcal{H}_k} \sum_{i=1}^{s} \frac{1}{\epsilon_i^2} \left( \phi(x_i)^\top \theta - y_i \right)^2 + \rho \|\theta\|_{\mathcal{H}_k}^2, \quad (2)$$

where $\epsilon_i = \|x_i\|_{V_{i-1}^{-1}}$ for $1 \leq i \leq s$, $\rho > 0$ is a regularizing parameter, and $V_s : \mathcal{H}_k \to \mathcal{H}_k$ is a positive-definite operator defined as

$$V_s = \rho I_s + \sum_{i=1}^{s} \frac{1}{\epsilon_i^2} \phi(x_i) \phi(x_i)^\top = \rho I_s + \Phi_s^\top W_s \Phi_s,$$

and $\Phi_s, Y_s$ and $W_s$ are defined as follows: $\Phi_s = (\phi(x_1), \phi(x_2), \ldots, \phi(x_s))^\top$, $Y_s = (y_1, y_2, \ldots, y_s)^\top$, $W_s = \operatorname{diag}\left(1/\epsilon_1^2, 1/\epsilon_2^2, \ldots, 1/\epsilon_s^2\right)$. Note that the above weighted least square estimator (2) can be represented as a closed-form, say, $\hat{\theta}_s = V_s^{-1} \Phi_s^\top W_s Y_s$. As in the linear case [Wan et al., 2023], the weighted least estimator is a key feature of the algorithm to achieve $O(\mathrm{poly}(\log T))$ regret bound (in the case of exponential eigendecay).

As previously mentioned, Algorithm 1 is a UCB-type algorithm and we need to compute an estimation $\widetilde{\mu}_s(x) := \phi(x)^\top \widehat{\theta}_s$ of $\mu(x)$ and an estimation error $\widetilde{\sigma}_s(x) := \|\phi(x)\|_{V_s^{-1}}$ for each $x \in \mathcal{X}$. However, naively, computation of $\widetilde{\mu}_s(x)$ and $\widetilde{\sigma}_s(x)$ requires computation of the linear operator $V_s^{-1}$ defined on $\mathcal{H}_k$, which is potentially infinite dimensional. It is well-known that in the unweighted (and classical) case [Valko et al., 2013, Srinivas et al., 2010], one can compute estimations of $\mu(x)$ and their estimation errors by using values of kernels and finite dimensional linear algebra (i.e., kernel trick) due to the reproducing property of the RKHS. The following proposition extends the well-known result to the weighted case.

**Proposition 4.1** (c.f. Dai et al. [2023], Sec. 4.1). *For $s \in \mathbb{Z}_{\geq 1}$ and $x \in \mathcal{X}$, we define $\widetilde{\mu}_s(x) = \phi(x)^\top \widehat{\theta}_s$ and $\widetilde{\sigma}_s(x) := \|\phi(x)\|_{V_s^{-1}}$. We also define a matrix $K_s \in \mathbb{R}^{s \times s}$ and a column vector $k_s(x) \in \mathbb{R}^s$ by $(K_s)_{ij} = (k(x_i, x_j))$ and $(k_s(x))_i = k(x, x_i)$ for $1 \leq i, j \leq s$. Then, we have the following.*

$$\widetilde{\mu}_s(x) = k_s(x)^\top (\rho I_s + W_s K_s)^{-1} W_s Y_s,$$
$$\rho \widetilde{\sigma}_s^2(x) = k(x, x) - k_s(x)^\top (\rho I_s + W_s K_s)^{-1} W_s k_s(x).$$

---

**Algorithm 1** QMCKernelUCB

---

**Inputs**: fail probability $\delta \in (0, 1)$, the total number of rounds $T$, an upper bound of the total number of stages $M$, and a tradeoff parameter $\eta > 0$.

1: **for** each stage $s = 1, 2, \ldots$ (terminate when we have used $T$ queries to all $\mathcal{O}_x, \mathcal{O}_x^\dagger$) **do**
2:      $x_s \leftarrow \operatorname{argmax}_{x \in \mathcal{X}} \widetilde{\mu}_{s-1}(x) + \beta_{s-1} \widetilde{\sigma}_{s-1}(x)$.
3:      $\epsilon_s \leftarrow \widetilde{\sigma}_{s-1}(x_s)$.
4:      Run QMC($\mathcal{O}_{x_s}, \eta \epsilon_s, \frac{\delta}{M}$) obtain an output $y_s$ of QMC.
5:      **for** the next $\frac{C}{\eta \epsilon_s} \log \frac{M}{\delta}$ rounds **do**
6:          play action $x_s$ and the player incurs regret $\mu(x^*) - \mu(x_s)$.
7:      **end for**
8: **end for**

---

## 4.2 CONFIDENCE INTERVAL

The following result provides a confidence interval of the estimation $\widetilde{\mu}_s(x)$.

**Proposition 4.2.** *Let $m$ be the total number of stage of Algorithm 1 and $x_s$ be the action selected by Algorithm 1 for each stage $s$. We assume that $M \geq m$, where $M$ is the parameter of Algorithm 1. With probability at least $1 - \delta$, the following inequality holds for any $s = 1, \ldots, m$ and $x \in \mathcal{X}$:*

$$|\mu(x) - \widetilde{\mu}_s(x)| \leq \beta_s \widetilde{\sigma}_s(x).$$

*Here, $\beta_s = \sqrt{\rho} S + \eta \sqrt{s}$ with $\|\theta\|_{\mathcal{H}_k} \leq S$.*

In the linear case, Wan et al. [2023] proved a similar result and in their result, $\beta_s$ is given as $O(\sqrt{ds})$, where $d$ is the dimension of the linear model. However, since $\dim \mathcal{H}_k$ is possibility infinite, their result is vacuous in our setting.

Although the proof is quite different, we note that Proposition 4.2 has some similarity to the known confidence interval in the classical setting [Srinivas et al., 2010]. In the classical setting, it is well-known that a confidence interval of the form $|\mu_t(x) - \mu(x)| = O(\sqrt{\gamma_T} \sigma_t(x))$ holds [Srinivas et al., 2010, Chowdhury and Gopalan, 2017], where $\mu_t(x)$ and $\sigma_t^2(x)$ are the posterior mean and positive variance in the classical setting, and $\gamma_T$ is the maximum information gain. By Proposition 4.2, we see that $|\mu(x) - \widetilde{\mu}_s(x)| = O(\sqrt{m} \widetilde{\sigma}(x))$ and as we shall see in Sec. 5, the total number $m$ of stages plays a similar role to the maximum information gain $\gamma_T$.

## 4.3 TRADEOFF PARAMETER

Both our algorithm (Algorithm 1) and Q-GP-UCB [Dai et al., 2023] are UCB-type algorithms that extend QLinUCB [Wan et al., 2023] to the kernelized case. However, we introduce a novel tradeoff parameter $\eta$ that tradeoffs the total

number of stages and regret incurred in each state. Since we call the reward oracles $O(\frac{1}{\eta \epsilon_s})$ times, if $\eta$ is larger, then regret incurred in each stage will be smaller, but the total number of stages will be larger. We detail the dependence of the parameter $\eta$ on the cumulative regret in Proposition 5.2.

# 5 REGRET ANALYSIS

In this section, we provide upper bounds of the cumulative regret of Algorithm 1. We present the main claims of the regret upper bounds in Sec. 5.1, and provide a proof sketch of the main result in Sec. 5.2. Recall that the complete proofs are provided in Appendix C.

## 5.1 MAIN RESULTS

Besides the assumptions introduced in Sec. 3, we make the following assumptions.

**Assumption 5.1.** (a) $k$ is a Mercer kernel, i.e., there exist a sequence of functions $\{\psi_i\}_{i \in I} \subset \mathcal{H}_k$ and positive numbers $\{\lambda_i\}_{i \in I}$ with $\lambda_1 \geq \lambda_2 \geq \cdots$ satisfying the statement of Theorem 3.1. (b) There exists a constant $\overline{\psi} > 0$ such that $\|\psi_i\|_\infty \leq \overline{\psi}$ for any $i \in I$. (c) There exists a constant $\overline{k} > 0$ such that $\sup_{x,x' \in \mathcal{X}} |k(x, x')| \leq \overline{k}$. (d) We assume that there exists $S > 0$ such that $\|\theta^*\|_{\mathcal{H}_k} \leq S$, where $\theta^*$ is the ground truth vector that determines the mean reward function $\mu$.

Here, assumptions (a), (b), (c) are assumed in the previous work [Vakili et al., 2021] and the assumption (d) is a boundedness condition, which is standard in the bandit literature.

We note that along with the standard bounded assumptions, we assume the reward function $\mu$ is normalized so that $\mu(x) \in [0, 1]$ for any $x \in \mathcal{X}$ in Section 3.2. Since the standard boundedness assumptions imply that the reward function is bounded [2], after normalization (or affine transformation) of rewards, the assumption $\mu(\mathcal{X}) \subseteq [0, 1]$ can be satisfied.

First, we introduce a regret upper bound of Algorithm 1 using the total number $m$ of stages. By Lemma 2.2, Proposition 4.2, and a standard proof technique for UCB-type algorithms, we can easily show that the cumulative regret of Algorithm 1 is bounded as follows:

**Proposition 5.2.** *Let $m$ be the total number of stages and, $\eta$ be a tradeoff parameter of Algorithm 1. We assume that $M \geq m$, where $M$ is the parameter of Algorithm 1. Then,*

---

[2] By standard the bounded assumptions, the Cauchy-Schwartz inequality, the reproducing property, we see that $|\mu(x)| \leq \langle \theta^*, \phi(x) \rangle \leq \|\theta^*\|_{\mathcal{H}_k} \|\phi(x)\|_{\mathcal{H}_k} \leq \|\theta^*\|_{\mathcal{H}_k} \sqrt{\overline{k}}$.

with probability at least $1 - \delta$, cumulative regret $R(T)$ of the algorithm is bounded by

$$R(T) = O\left(m(\eta^{-1} + \sqrt{m})\log(M/\delta)\right).$$

We remark that the total number $m$ depends on $\eta$. If we take a large $\eta$, then the number of oracle queries by the QMC method will be smaller, and the total number $m$ of stages will be larger. By Proposition 5.2, we have to appropriately select $\eta$ and provide an upper bound of $m$.

The following proposition provides upper bounds of $m$.

**Proposition 5.3.** *Assume $T > 1, M \geq e$ and let $m$ be the total number of stages of Algorithm 1.*

1. *Suppose that $k$ has a $(C_p, \beta_p)$ polynomial eigendecay. We take $\eta$ as*

$$\eta = T^{-\frac{1}{1+\beta_p}}.$$

   *Then, there exists a constant $c_p > 0$ depending only on $C_p, \beta_p, \rho, \overline{k}, \overline{\psi}$ satisfying the following inequality:*

$$m \leq c_p T^{\frac{2}{1+\beta_p}} \log^{1-\beta_p^{-1}}(T).$$

2. *Suppose that $k$ has a $(C_{e,1}, C_{e,2}, \beta_e)$ exponential eigendecay. We take $\eta = 1$. Then, there exists a constant $c_e > 0$ depending only on $C_{e,1}, C_{e,2}, \beta_e, \rho, \overline{k}, \overline{\psi}$ satisfying the following inequality:*

$$m \leq c_e \log^{1+1/\beta_e}(T).$$

In Proposition 5.3, in the case of the polynomial eigendecay, we select $\eta$ so that $\eta^{-1}$ and the upper bound of $\sqrt{m}$ have the same order of $T$ and in the case of the exponential decay, we select $\eta = 1$. We note that upper bounds provided in Proposition 5.3 have similarity to upper bounds of the maximum information gain $\gamma_T$ [Vakili et al., 2021, Corollary 1]. Actually, Dai et al. [2023] showed that the total number $m$ of stages with the tradeoff parameter $\eta = 1$ has the same bound as $\gamma_{T^2}$. Due to the appropriate choice of the tradeoff parameter, our results (Proposition 5.3) improves their result $\widetilde{O}(T^{2/\beta_p})$ in the case of the polynomial eigendecay.

Therefore, by Proposition 5.3 and Proposition 5.2, we obtain the following theorem, which is the main result of this paper.

**Theorem 5.4** (Upper Bounds of Algorithm 1). *Assume $T > 1$. Suppose that Assumption 5.1 holds.*

1. *Suppose that the kernel $k$ has a $\beta_p$ polynomial eigendecay. Let $\eta$ and $c_p$ be as in Proposition 5.3. Then, with probability at least $1 - \delta$, the cumulative regret of Algorithm 1 with $M = c_p \eta^{-2}$ is bounded as*

$$R(T) = O\left(T^{\frac{3}{1+\beta_p}} \log^{3(1-\beta_p^{-1})/2}(T) \log\left(\frac{T}{\delta}\right)\right).$$

2. *Suppose that the kernel $k$ has a $\beta_e$ exponential eigendecay. Then with probability at least $1 - \delta$, the cumulative regret of Algorithm 1 with $M = c_e \log^{1+1/\beta_e}(T)$ and $\eta = 1$ is bounded as*

$$R(T) = O\left(\log^{3(1+\beta_e^{-1})/2}(T) \log\left(\frac{\log T}{\delta}\right)\right),$$

*where $c_e$ is the constant provided in Proposition 5.3.*

Theorem 5.4 indicates that our regret upper bound exponentially improves that of the classical algorithms [Valko et al., 2013, Vakili et al., 2021] if $k$ has an exponential eigendecay. Moreover, our regret upper bound is better than the classical bounds if a polynomial eigendecay with large $\beta_p$. Moreover, we note that our regret upper bound improves that of [Dai et al., 2023] in the case of polynomial eigendecay. More precisely, in the case of a $\beta_p$-polynomial eigendecay, while the regret upper bound of [Dai et al., 2023] is given as $\widetilde{O}\left(T^{\frac{3}{\beta_p}} \log\left(\frac{1}{\delta}\right)\right)$, that of ours is $\widetilde{O}\left(T^{\frac{3}{1+\beta_p}} \log\left(\frac{1}{\delta}\right)\right)$. We also note that our regret bound is better than the regret bound $\widetilde{O}(T^{1/2+1/\beta_p})$ of GP-UCB [Srinivas et al., 2010] whenever GP-UCB has sublinear regret (i.e., $\beta_p > 2$), while that of Q-GP-UCB [Dai et al., 2023] is not necessarily better than GP-UCB.

## 5.2 SKETCH OF THE PROOF

As previously mentioned, providing the upper bounds of the total number of stages $m$ is a key step to prove the main result Theorem 5.4. In this section, we provide a sketch of the proof of Proposition 5.3. Following [Wan et al., 2023], we first relate $m$ to the log-determinant $\gamma^{\mathrm{QMC}}$ of the positive operator $\rho^{-1} V_m$. Then, by considering a projection onto a finite dimensional subspace of $\mathcal{H}_k$, we provide an upper bound of $\gamma^{\mathrm{QMC}}$.

Let $K_m, W_m \in \mathbb{R}^{m \times m}$ be matrices and $\mathcal{Q}_m : \mathcal{H}_k \to \mathcal{H}_k$ be the positive semi-definite operator defined in Sec. 4.

$$\mathcal{Q}_m = \sum_{s=1}^{m} \epsilon_s^{-2} \phi(x_s)\phi(x_s)^{\top}.$$

We define $\gamma^{\mathrm{QMC}} > 0$ by

$$\log\det\left(I_m + \rho^{-1} K'\right) = \log\det\left(I_{\mathcal{H}_k} + \rho^{-1}\mathcal{Q}_m\right), \quad (3)$$

where $K' = W_m^{1/2} K_m W_m^{1/2}$. We note that (3) holds by the Weinstein-Aronszajn identity. If the weight matrix is the identity matrix, then the definition of $\gamma^{\mathrm{QMC}}$ is almost identical to that of the maximum information gain $\gamma_T$ defined as $\gamma_T = \sup_{\xi_1,\ldots,\xi_T \in \mathcal{X}} \log\det\left(I_T + K(\boldsymbol{\xi})\right)$, where $K(\boldsymbol{\xi}) = (k(\xi_i, \xi_j))_{1 \leq i,j \leq T} \in \mathbb{R}^{T \times T}$. However, unlike $\gamma_T$, $\gamma^{\mathrm{QMC}}$ depends on the matrix size $m$ trivially. More precisely, it can be proved that $\gamma^{\mathrm{QMC}} = m \log 2$ (Lemma C.5). Therefore, to bound $m$, it is sufficient to bound $\gamma^{\mathrm{QMC}}$.

If the RKHS $\mathcal{H}_k$ is finite dimensional, [Wan et al., 2023, Lemma 2] provides an upper bound of $\gamma^{\mathrm{QMC}}$ of the form $O(\dim \mathcal{H}_k \log(T))$. However, this bound is vacuous since $\dim \mathcal{H}_k$ can be infinite. In an attempt of deriving an upper bound of the maximum information gain $\gamma_T$, there was a similar issue. Vakili et al. [2021] resolved the issue by considering a projection of $\mathcal{H}_k$ to a finite dimensional subspace and we take a similar approach.

We recall that a set of functions $\{\lambda_i^{1/2}\psi_i\}_{i\in I}$ forms an orthonormal basis of the RKHS $\mathcal{H}_k$ (Theorem 3.1), where $I = \{1, 2, \cdots, \dim \mathcal{H}_k\}$ if $\mathcal{H}_k$ is finite dimensional and $I = \mathbb{Z}_{\geq 1}$ otherwise. For a positive integer $D$, we define an orthogonal projection $\mathcal{P}_D : \mathcal{H}_k \to \mathcal{H}_k$ by $f \mapsto \sum_{i=1}^{D} \langle f, \lambda_i^{1/2}\psi_i \rangle_{\mathcal{H}_k} \lambda_i^{1/2}\psi_i$. Then, $\mathcal{P}_D(f)$ gives an approximation of $f$ in the finite dimensional subspace $\mathcal{P}_D(\mathcal{H}_k)$. To bound $\gamma^{\mathrm{QMC}}$, one can mimic the proof of [Vakili et al., 2021, Theorem 3], however, we provide a more generalized result. Below, we show that our upper bounds of $\gamma^{\mathrm{QMC}}$ (Corollary 5.6) can be derived from the following proposition. We also note that the proof of the following proposition provides a simple alternative proof of [Vakili et al., 2021, Theorem 3].

**Proposition 5.5.** *Let $\pi : \mathcal{H}_k \to \mathcal{H}_k$ be a projection operator of finite rank and $U : \mathcal{H}_k \to \mathcal{H}_k$ be a positive semi-definite operator of finite-rank. We assume that the range (image) $\mathrm{Ran}\,\pi$ of $\pi$ is $D$-dimensional with $D < \infty$. Then, the following inequality holds:*

$$\log \det(I + U) \leq D \log \left(1 + \frac{\mathrm{Tr}\,U\pi}{D}\right) + \mathrm{Tr}\,U(I - \pi).$$

We apply Proposition 5.5 to the case when $\pi = \mathcal{P}_D, U = \rho^{-1}\mathcal{Q}_m$. We can bound $\mathrm{Tr}\,\mathcal{Q}_m\mathcal{P}_D$ by $\mathrm{Tr}\,\mathcal{Q}_m\mathcal{P}_D \leq \mathrm{Tr}\,\mathcal{Q}_m = \mathrm{Tr}\,W_m^{1/2}K_m W_m^{1/2} \leq \overline{k}\,\mathrm{Tr}\,W_m$. To compute $\mathrm{Tr}\,\mathcal{Q}_m(I - \mathcal{P}_D)$, it is sufficient to compute $t(i, x) := \lambda_i^{1/2}\psi_i^{\top}\phi(x)\phi(x)^{\top}\lambda_i^{1/2}\psi_i$ for each $i \in I$ and $x \in \mathcal{X}$. By the reproducing property, we have $t(i, x) = \lambda_i\psi_i^2(x) \leq \lambda_i\overline{\psi}^2$. Therefore,

$$\mathrm{Tr}\,\mathcal{Q}_m(I - \mathcal{P}_D) = \sum_{i > D}\sum_{s=1}^{m}\epsilon_s^{-2}t(i, x_s)$$
$$\leq \sum_{i > D}\lambda_i\overline{\psi}^2\sum_{s=1}^{m}\epsilon_s^{-2} = \delta_D\,\mathrm{Tr}\,W_m,$$

where $\delta_D$ is defined as $\sum_{i \in I, i > D}\lambda_i\overline{\psi}^2$. Thus, we obtain the following.

**Corollary 5.6.** *We define $E$ by $\sum_{s=1}^{m}\epsilon_s^{-2}$, i.e., $E = \mathrm{Tr}\,W_m$. Then, for any $D \in \mathbb{Z}_{\geq 1}$, the following inequality holds:*

$$\gamma^{\mathrm{QMC}} \leq D \log \left(1 + \frac{\overline{k}}{D\rho}E\right) + \frac{\delta_D}{\rho}E.$$

Then, by Corollary 5.6 and using the same argument as the proof of [Vakili et al., 2021, Corollary 1], we can bound $\gamma^{\mathrm{QMC}}$ in terms of $E$. Since the total number of oracle queries of Algorithm 1 is limited up to $T$, we have $T \gtrsim \eta^{-1}\sum_{i=1}^{m}\epsilon_i^{-1} \geq \sqrt{E}$. Therefore, we can bound the $\gamma^{\mathrm{QMC}}$ in terms of $\eta T$. By selecting the $\eta^{-1}$ as the same order as an upper bound of $\gamma^{\mathrm{QMC}}$, we can provide an upper bound of $\gamma^{\mathrm{QMC}}$ in terms of $T$. By $\gamma^{\mathrm{QMC}} = m \log 2$ (Lemma C.5), we have the assertion of Proposition 5.3.

# 6 LITERATURE OVERVIEW

This section consists of two parts: a comparison of this study with Dai et al. [2023], and a review of existing work on kernelized bandits and quantum online learning algorithms.

## 6.1 COMPARISON TO QUANTUM BAYESIAN OPTIMIZATION

### 6.1.1 Bias of the QMC Estimator

As stated in Section 1, the present paper deals with the same problem setting as Dai et al. [2023]. In particular, Dai et al. [2023] provided a similar confidence interval to Proposition 4.2 under the following assumption. We discuss the validity of this assumption below.

**Assumption 6.1** (Subgaussian Error Assumption of QMC). Let $y$ be a random variable taking values in $[0, 1]$ and $\mathcal{O}(y)$ the unitary operator corresponding to $y$ as in Lemma 2.2. Let $\widehat{y}$ be an output of the QMC method $\mathrm{QMC}(\mathcal{O}(y), \epsilon, \delta)$ introduced in Lemma 2.2. Then, the error $y - \widehat{y}$ is $\epsilon$-subgaussian.

Dai et al. [2023] claimed that this assumption is assured by Lemma 2.2, however, Lemma 2.2 only states that $|y - \widehat{y}|$ is bounded by $\epsilon$ with a high probability. Noting that the subgaussian property implies that the error $y - \widehat{y}$ is unbiased, their argument implies the QMC estimator is unbiased.

An implementation of the QMC method calls the quantum phase estimation algorithm repeatedly, obtains estimated phases $\widehat{\Theta}_1, \ldots, \widehat{\Theta}_n \in [0, 2\pi]$, computes a median $\widehat{\Theta} = \mathrm{Median}(\widehat{\Theta}_1, \ldots, \widehat{\Theta}_n)$, and outputs an estimation $(1 - \cos(\widehat{\Theta}/2))/2$ of $\mathbb{E}[y]$ (c.f., Rebentrost et al. [2018]). Since each phase estimation $\widehat{\Theta}_i$ includes an approximation error due to a finite number of qubits, and the function $(1 - \cos(x))/2$ is non-linear, to the best of our knowledge, there is no evidence that indicates the QMC estimator is unbiased.

Although there are some recent methods for mitigating the bias of the QMC (or Quantum Amplitude) estimator, to the best of our understanding, these improved methods are still biased and require a larger number of oracle queries (see [Miyamoto, 2023] or references therein).

### 6.1.2 Improved Regret Bounds

As we discussed in the introduction and the remark after Theorem 5.4, in the case of polynomial eigendecay our regret bound improves that of [Dai et al., 2023] even under the unbiasedness assumption of the QMC estimator.

### 6.1.3 Tradeoff Parameter

Both our algorithm (Algorithm 1) and Q-GP-UCB [Dai et al., 2023] are UCB-type algorithms that extend QLinUCB [Wan et al., 2023] to the kernelized case. As we discussed in Section 4.3, we introduced a novel parameter $\eta$ that tradeoffs the total number of stages and regret incurred in each state, which provides aforementioned improved regret bounds.

### 6.1.4 Proof Technique for Bounding the (weighted) Information Gain

We note that [Vakili et al., 2021, Theorem 3] can be derived by Proposition 5.5 by a similar argument when deriving Corollary 5.6. In particular, the proof of Proposition 5.5 and analysis provided Section 5.2 provide a simple alternative proof to [Vakili et al., 2021, Theorem 3]. Although Dai et al. [2023] proved the same result as Corollary 5.6, we can say Proposition 5.5 is a more general since both [Vakili et al., 2021, Theorem 3] and Corollary 5.6 are corollaries of this proposition.

### 6.2 RELATED WORK

In the classical setting, Valko et al. [2013] discussed a kernelized UCB algorithm as an a non-linear extension of LinUCB [Li et al., 2010], and provided a cumulative regret bound based on a notion of the effective dimension. The effective dimension of an RKHS is essential same as the information gain [Vakili et al., 2021, Remark 1]. Vakili et al. [2021] derived general upper bounds of the information gain under conditions on the eigendecay of the kernel.

As for the prior works on bandit problems in the quantum setting, Wan et al. [2023] studied a quantum multi-armed bandits and stochastic linear bandits with linear reward model and introduced a quantum algorithm that enjoys quadratic speedup compared to the best possible classical result. Dai et al. [2023] extended the work [Wan et al., 2023] to the case of a non-linear reward model and proposed a similar algorithm based on kernelization under the unbiasedness assumption of the QMC estimator. Besides these studies, Li and Zhan [2022] studied a quantum bandit convex optimization problem and Wang et al. [2021] studied a best arm identification problem in the quantum multi-arm bandit setting. The algorithms proposed in these studies are also stage-based as in the present study and have been shown to achieve a quantum speedup compared to the classical

algorithms. However, these algorithms are quite different from ours due to the different problem settings.

## 7 CONCLUSION

In this paper, we proposed a UCB-type algorithm for quantum bandit problems where the reward function is non-linear with respect to an action. By employing Mercer's theorem, we provided a theoretical analysis that the proposed algorithm achieves $O(\text{poly}(\log T))$ regret bound when the decay rate of Mercer operator decreases exponentially fast. A limitation of this study is that the proposed method calls a Quantum Monte Carlo method in each round, which would require waiting for the advent of a fault-tolerant quantum computation. For future research direction, it would be intriguing to investigate the possibility of designing an algorithm that does not necessitate the computation of matrix inversion, such as Langevin Monte Carlo Thompson Sampling (LMC-TS) [Xu et al., 2022] which is based on noisy gradient descent updates. Moreover, the optimality of our algorithm remains unknown, and thus exploring a lower bound of the cumulative regret in this problem setting is an important open problem.

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

# Supplementary Material

**Yasunari Hikima**[*1]          **Kazunori Murao**[*1]          **Sho Takemori**[*1]          **Yuhei Umeda**[*1]

[1]AI Laboratory, Fujitsu Limited, Kawasaki, Japan

## APPENDIX

## A   NOTATION

We list the notation that is used in this paper in Table 2

| Symbol | Meaning |
|---|---|
| $\mathcal{V}$ | a finite-dimensional Hilbert space over $\mathbb{C}$ |
| $|x\rangle$ | a state vector in $\mathcal{V}$ |
| $\langle x|$ | a corresponding state vector $|x\rangle^{\dagger}$ in the dual space of $\mathcal{V}$ |
| $M_{\sigma} : \mathcal{V} \to \mathcal{V}$ | a measurement operator |
| $k : \mathcal{X} \times \mathcal{X} \to \mathbb{R}$ | a semi-positive definite kernel |
| $\mathcal{H}_k$ | a RKHS corresponding to the kernel $k$ |
| $\mathcal{X}$ | a set of actions |
| $\mu : \mathcal{X} \to [0, 1]$ | a mean reward function |
| $\mathcal{V}_n = (\mathbb{C}^2)^{\otimes n}$ | a state space of $n$ qubits |
| $\Lambda_n$ | a set of binary sequences of length $n$ |
| $T$ | a time horizon |
| $\phi : \mathcal{X} \to \mathcal{H}_k$ | a feature map |
| $m$ | a maximum stage of our algorithm |
| $M$ | an input parameter of Algorithm 1 |
| $\mathcal{G}_s$ | a finite-dimensional subspace of $\mathcal{H}_k$ spanned by $\phi(x_1), \ldots, \phi(x_s)$ |
| $\| \cdot \|_p$ | $\ell_p$ norm ($1 \leq p \leq \infty$) |
| $\| \cdot \|_{\mathrm{F}}$ | Frobenius norm of a matrix |
| $\sigma_{\max}(\cdot)$ | the spectral norm of a matrix |
| $A|_{\mathcal{G}}$ | the restriction of an operator $A$ on a domain $\mathcal{G}$ |
| $\mathrm{Ran}\, A$ | the range (image) of an operator $A$ |
| $\rho$ | a regularizing parameter |
| $\{\lambda_i\}_{i \in I}$ | eigenvalues of the Mercer operator $\mathcal{T}_k$ |
| $\{\psi_i\}_{i \in I}$ | eigenfunctions of the Mercer operator $\mathcal{T}_k$ |

Table 2: Notation

# B   REMARKS ON OPERATOR DETERMINANT AND TRACE

In Sec. 5.2, we consider the log determinant of the operator $I_{\mathcal{H}_k} + \rho^{-1}\mathcal{Q}_m : \mathcal{H}_k \to \mathcal{H}_k$. In this section, we note that there is no difficulty in defining the determinant since $\mathcal{Q}_m$ is a finite-rank operator. In this paper, we define determinants and traces of a specific form of operators as follows.

**Definition B.1.** Let $A : \mathcal{H}_k \to \mathcal{H}_k$ be a finite rank operator. We define the determinant $\det(I_{\mathcal{H}_k} + A)$ and trace $\operatorname{Tr} A$ as

$$\det(I_{\mathcal{H}_k} + A) = \det(I_{\mathcal{G}} + A|_{\mathcal{G}}), \quad \operatorname{Tr} A = \operatorname{Tr} A|_{\mathcal{G}},$$

where $\mathcal{G}$ is a finite dimensional subspace of $\mathcal{H}_k$ such that $\operatorname{Ran} A \subseteq \mathcal{G}$, and $A|_{\mathcal{G}}$ denotes the restriction of $A$ on $\mathcal{G}$. Moreover, these definitions do not depend on the choices of $\mathcal{G}$ satisfying the same property.

To show the last statement, it is sufficient to prove that $\det(I_{\mathcal{G}} + A|_{\mathcal{G}}) = \det(I_{\operatorname{Ran} A} + A|_{\operatorname{Ran} A})$ and $\operatorname{Tr} A|_{\mathcal{G}} = \operatorname{Tr} A|_{\operatorname{Ran} A}$, but they can be easily verified by considering the matrix representation of $A|_{\mathcal{G}}$ in the finite dimensional subspace $\mathcal{G}$.

We also remark that in this special case, the above determinant is identical to the Fredholm determinant [Simon, 2005], which can be defined for the trace class. However, we only consider $\det(I_{\mathcal{H}_k} + A)$ with a finite-rank operator $A$, the above definition suffices for our purpose.

# C   PROOFS

## C.1   PROOF OF PROPOSITION C.1

We also refer to [Dai et al., 2023, Appendix A] for the proof.

To emphasize Proposition C.1 holds for any weights and sequence $x_1, x_2, \cdots$, we prove the following.

**Proposition C.1.** *Let $n \in \mathbb{Z}_{\geq 1}$, $x_1, \ldots, x_n \in \mathcal{X}$, $\omega_1, \ldots, \omega_n > 0$, $y_1, \ldots, y_n \in \mathbb{R}$. Define a self-adjoint operator $\mathcal{S} : \mathcal{H}_k \to \mathcal{H}_k$ as $\mathcal{S} = \rho 1 + \sum_{i=1}^{n} \omega_i \phi(x_i)\phi(x_i)^{\top}$. We put $Y_n = (y_1, \cdots, y_n)^{\top} \in \mathbb{R}^n$, $\boldsymbol{\omega} = \operatorname{diag}(\omega_1, \ldots, \omega_n)$, and define $K \in \mathbb{R}^{n \times n}$ as $K_{ij} = k(x_i, x_j)$. Then for any $x \in \mathcal{X}$, we have*

$$\phi(x)^{\top}\mathcal{S}^{-1}\sum_{i=1}^{n}\omega_i y_i \phi(x_i) = k(x)^{\top}(\rho I_n + \boldsymbol{\omega}K)^{-1}\boldsymbol{\omega}Y_n,$$

$$\rho\|\phi(x)\|_{\mathcal{S}^{-1}}^2 = k(x,x) - k(x)^{\top}(\rho I_n + \boldsymbol{\omega}K)^{-1}\boldsymbol{\omega}k(x).$$

*Here, $k(x) \in \mathbb{R}^n$ is a column vector defined by $(k(x))_i = k(x, x_i)$ for $1 \leq i \leq n$.*

*Proof.* Let $x \in \mathcal{X}$. Since for $1 \leq i \leq n$,

$$\mathcal{S}\phi(x_i) = \rho\phi(x_i) + \sum_j \omega_j k(x_i, x_j)\phi(x_j) = \sum_j (\rho\delta_{ij} + \omega_j k(x_i, x_j))\phi(x_j),$$

we have

$$(\mathcal{S}\phi(x_1), \cdots, \mathcal{S}\phi(x_n)) = (\phi(x_1), \cdots, \phi(x_n))(\rho I_n + \boldsymbol{\omega}K_n).$$

Thus, we see that $\mathcal{S}\sum_i a_i \phi(x_i) = \sum_i \omega_i y_i \phi(x_i)$, where $a \in \mathbb{R}^n$ is given as $(\rho I_n + \boldsymbol{\omega}K)^{-1}\boldsymbol{\omega}Y_n$. Therefore, we have the first statement of the proposition by the reproducing property.

Similarly, we have the following equality:

$$(\mathcal{S}\phi(x_1), \cdots, \mathcal{S}\phi(x_n), \mathcal{S}\phi(x)) = (\phi(x_1), \cdots, \phi(x_n), \phi(x))\begin{pmatrix} \rho I_n + \boldsymbol{\omega}K & \boldsymbol{\omega}k(x) \\ 0 & \rho \end{pmatrix}.$$

Define $b \in \mathbb{R}^n$ by $(\rho I_n + \boldsymbol{\omega}K)^{-1}\boldsymbol{\omega}k(x)$. Then,

$$(\mathcal{S}\phi(x_1), \cdots, \mathcal{S}\phi(x_n), \mathcal{S}\phi(x))\begin{pmatrix} -b \\ 1 \end{pmatrix} = (\phi(x_1), \cdots, \phi(x_n), \phi(x))\begin{pmatrix} \rho I_n + \boldsymbol{\omega}K & \boldsymbol{\omega}k(x) \\ 0 & \rho \end{pmatrix}\begin{pmatrix} -b \\ 1 \end{pmatrix}$$

$$= \rho\phi(x).$$

Thus, it follows that

$$\mathcal{S}\left(\phi(x) - \sum_{i=1}^{n} b_i \phi(x_i)\right) = \rho\phi(x),$$

Therefore, we have the following.

$$\rho\|\phi(x)\|_{\mathcal{S}^{-1}}^2 = \rho\phi(x)^\top \mathcal{S}^{-1}\phi(x) = \phi(x)^\top\left(\phi(x) - \sum_{i=1}^{n} b_i\phi(x_i)\right) = k(x,x) - k(x)^\top b.$$

Here, the last equality holds from the reproducing property. This completes the proof. $\qquad\square$

## C.2  PROOF OF PROPOSITION 4.2

For any $x \in \mathcal{X}$, we have

$$|\mu(x) - \widetilde{\mu}_s(x)| = |\phi(x)^\top\theta^* - \phi(x)^\top\widehat{\theta}_s| \le \|\phi(x)\|_{V_s^{-1}}\left\|\theta^* - \widehat{\theta}_s\right\|_{V_s} = \left\|\theta^* - \widehat{\theta}_s\right\|_{V_s}\widetilde{\sigma}_s(x).$$

Thus, Proposition 4.2 follows from the following lemma.

**Lemma C.2.** *Let $m$ be the total number of stage of Algorithm 1 and assume that $M \ge m$. Then, with probability at least $1 - \delta$, the following inequality holds for any $s = 1, \ldots, m$:*

$$\left\|\theta^* - \widehat{\theta}_s\right\|_{V_s} \le \beta_s.$$

*Here, $\beta_s = \sqrt{\rho}S + \eta\sqrt{s}$.*

*Proof.* This can be proved similarly to [Wan et al., 2023, Lemma 3]. However, since a naive application of their proof would lead to a bound involving the dimension of the RHKS $\mathcal{H}_k$, we bound $\left\|\theta^* - \widehat{\theta}_s\right\|_{V_s}$ as follows.

Define an event $\mathcal{E}$ as

$$\mathcal{E} = \{|\mu(x_s) - y_s| \le \varepsilon_s, \quad 1 \le s \le m\}.$$

Since there are $m$ stages in Algorithm 1 and $M \ge m$, by taking a union bound,

$$P(\mathcal{E}) \ge 1 - \frac{m}{M}\delta \ge 1 - \delta.$$

In the rest of the proof, we assume that the event $\mathcal{E}$ holds. By the same proof as [Wan et al., 2023, Lemma 3], i.e., by replacing $x_s$ by $\phi(x_s)$ in their proof, with probability at least $1 - \delta$, we have

$$\|\theta^* - \widehat{\theta}_s\|_{V_s} \le \rho\|\theta^*\|_{V_s^{-1}} + \|\Phi_s^\top W_s^{1/2}\Gamma_s\|_{V_s^{-1}}. \tag{4}$$

Here, $\Gamma_s = W_s^{1/2}(\Phi_s\theta^* - Y_s) \in \mathbb{R}^s$. Then, by Lemma 2.2, we have $\|\Gamma_s\|_\infty \le \eta$. We note that $\Phi_s$ is a linear operator from $\mathcal{H}_k$ to $\mathbb{R}^s$ defined by $\mathcal{H}_k \ni f \mapsto (\phi(x_1)^\top f, \cdots, \phi(x_s)^\top f) = (f(x_1), \cdots, f(x_s)) \in \mathbb{R}^s$. Similarly, $\Phi_s^\top$ is a linear operator from $\mathbb{R}^s$ to $\mathcal{H}_k$ defined by $(a_i)_{1 \le i \le s} \mapsto \sum_{i=1}^s a_i\phi(x_i)$. Therefore, we note that $\Phi_s V_s^{-1}\Phi_s^\top$ is a linear operator from $\mathbb{R}^s \to \mathbb{R}^s$, i.e., a matrix in $\mathbb{R}^{s\times s}$. Since the trace norm is dual to the spectral norm with respect to the inner product of the space of symmetric matrices defined as $(A, B) \mapsto \operatorname{Tr} AB$, we have

$$\begin{aligned}
\|\Phi_s^\top W_s^{1/2}\Gamma_s\|_{V_s^{-1}}^2 &= \Gamma_s^\top W_s^{1/2}\Phi_s V_s^{-1}\Phi_s^\top W_s^{1/2}\Gamma_s \\
&= \operatorname{Tr}\Gamma_s\Gamma_s^\top W_s^{1/2}\Phi_s V_s^{-1}\Phi_s^\top W_s^{1/2} \\
&\le \operatorname{Tr}\left(\Gamma_s\Gamma_s^\top\right)\sigma_{\max}\left(W_s^{1/2}\Phi_s V_s^{-1}\Phi_s^\top W_s^{1/2}\right) \\
&\le \eta^2 s\,\sigma_{\max}\left(W_s^{1/2}\Phi_s V_s^{-1}\Phi_s^\top W_s^{1/2}\right).
\end{aligned}$$

Here, $\sigma_{\max}$ denotes the spectral norm and the last inequality follows from $\|\Gamma_s\|_\infty \le \eta$. We put $\widetilde{\Phi}_s = W_s^{1/2}\Phi_s$. Then, noting that

$$(\rho I + \widetilde{\Phi}_s^\top\widetilde{\Phi}_s)\widetilde{\Phi}_s^\top = \widetilde{\Phi}_s^\top(\rho I + \widetilde{\Phi}_s\widetilde{\Phi}_s^\top),$$

we have the following (c.f., Valko et al. [2013]):

$$\widetilde{\Phi}_s^\top (\rho I + \widetilde{\Phi}_s \widetilde{\Phi}_s^\top)^{-1} = (\rho I + \widetilde{\Phi}_s^\top \widetilde{\Phi}_s)^{-1} \widetilde{\Phi}_s^\top.$$

Thus, we have

$$\sigma_{\max}\left(W_s^{1/2}\Phi_s V_s^{-1}\Phi_s^\top W_s^{1/2}\right) = \sigma_{\max}\left(\widetilde{\Phi}_s \left(\rho I + \widetilde{\Phi}_s^\top \widetilde{\Phi}_s\right)^{-1}\widetilde{\Phi}_s^\top\right) = \sigma_{\max}\left(\widetilde{\Phi}_s \widetilde{\Phi}_s^\top (\rho I + \widetilde{\Phi}_s \widetilde{\Phi}_s^\top)^{-1}\right) \leq 1.$$

Therefore, we see that $\|\Phi_s^\top W_s^{1/2}\Gamma_s\|_{V_s^{-1}} \leq \eta\sqrt{s}$. Noting that $\|\theta^*\|_{V_s^{-1}} \leq \rho^{-1/2}\|\theta^*\|_{\mathcal{H}_k}$, we have our assertion by (4). $\quad\square$

## C.3  PROOF OF PROPOSITION 5.2

*Proof.* This can be proved by a standard argument for the analysis of UCB-type algorithm, Lemma 2.2, and Proposition 4.2. We let $\mathcal{E}$ be an event on which the inequalities in Proposition 4.2 hold and assume that $\mathcal{E}$ holds. Let $x^* = \mathrm{argmax}_{x \in \mathcal{X}} \phi(x)^\top \theta^*$. By definition of $x_s$ and Proposition 4.2, we have

$$\mu(x^*) - \mu(x_s) \leq \widetilde{\mu}_{s-1}(x^*) + \beta_{s-1}\widetilde{\sigma}_{s-1}(x^*) - \widetilde{\mu}_{s-1}(x_s) + \beta_{s-1}\widetilde{\sigma}_{s-1}(x_s)$$
$$\leq 2\beta_{s-1}\widetilde{\sigma}_{s-1}(x_s) = 2\beta_{s-1}\epsilon_s.$$

Thus, by Lemma 2.2, the regret that the player incurs in the stage $s$ is at most

$$2\beta_{s-1}\epsilon_s C \log(M/\delta)\frac{1}{\eta\epsilon_s} = 2C(\sqrt{\rho}S\eta^{-1} + \sqrt{s-1})\log(M/\delta) \leq 2C(\sqrt{\rho}S\eta^{-1} + \sqrt{m})\log(M/\delta).$$

Therefore, with probability at least $1 - \delta$, the cumulative regret $R(T)$ is bounded as

$$R(T) \leq 2Cm(\sqrt{\rho}S\eta^{-1} + \sqrt{m})\log(M/\delta) = O(m(\eta^{-1} + \sqrt{m})\log(M/\delta)).$$

$\quad\square$

## C.4  PROOF OF PROPOSITION 5.5

*Proof.* As we remarked in Sec. B, we note that there is no difficulty in defining $\log\det(I + U)$ since $U$ is of finite-rank. We decompose $\mathcal{H}_k$ as $\mathcal{H}_k = \mathcal{H}_1 \oplus \mathcal{H}_2$ by the projection $\pi$, where $\mathcal{H}_1 = \mathrm{Ran}\,\pi$ and $\mathcal{H}_2 = \mathrm{Ran}(I - \pi)$. We define $U_{11} : \mathcal{H}_1 \to \mathcal{H}_1$, $U_{22} : \mathcal{H}_2 \to \mathcal{H}_2$, and $U_{12} : \mathcal{H}_2 \to \mathcal{H}_1$ by $U_{11} = \pi U|_{\mathcal{H}_1}$, $U_{22} = (I - \pi)U|_{\mathcal{H}_2}$, $U_{12} = (I - \pi)U|_{\mathcal{H}_2}$. That is, with respect to the decomposition $\mathcal{H}_1 \oplus \mathcal{H}_2$, $U$ can be represented by a matrix $\begin{pmatrix} U_{11} & U_{12} \\ U_{12}^\top & U_{22} \end{pmatrix}$. Since

$$\begin{pmatrix} I & 0 \\ -U_{12}^\top(I + U_{11})^{-1} & I \end{pmatrix}\begin{pmatrix} I + U_{11} & U_{12} \\ U_{12}^\top & I + U_{22} \end{pmatrix}\begin{pmatrix} I & -(I + U_{11})^{-1}U_{12} \\ 0 & I \end{pmatrix}$$
$$= \begin{pmatrix} I + U_{11} & 0 \\ 0 & I + U_{22} - U_{12}^\top(I + U_{11})^{-1}U_{12} \end{pmatrix}$$

and noting that they are finite-rank operators, we have

$$\log\det(I + U) = \log\det(I + U_{11}) + \log\det(I + U_{22} - U_{12}^\top(I + U_{11})^{-1}U_{12}).$$

We introduce the following well-known matrix inequalities (for the first inequality, we refer to [Vakili et al., 2021, Lemma 1]). For a positive semi-definite matrix $A \in \mathbb{R}^{n \times n}$, we have

$$\log\det(I + A) \leq n\log(1 + \mathrm{Tr}\,A/n), \quad \log\det(1 + A) \leq \mathrm{Tr}\,A.$$

Noting that $\dim\mathcal{H}_1 = D$ and these matrix inequalities hold for finite rank operators, we see that

$$\log\det(I + U) \leq D\log(1 + \mathrm{Tr}\,U_{11}/D) + \mathrm{Tr}\left(U_{22} - U_{12}^\top(I + U_{11})^{-1}U_{12}\right)$$
$$\leq D\log(1 + \mathrm{Tr}\,U_{11}/D) + \mathrm{Tr}\,U_{22}.$$

Here, the second inequality holds since $U_{12}^\top(I + U_{11})^{-1}U_{12}$ is positive semi-definite. We have our assertion by noting that $\mathrm{Tr}\,U_{11} = \mathrm{Tr}\,U\pi$ and $\mathrm{Tr}\,U_{22} = \mathrm{Tr}\,U(I - \pi)$. $\quad\square$

## C.5 PROOF OF PROPOSITION 5.3

To prove Proposition 5.3, first, we relate $\sum_{s=1}^{m} \frac{1}{\epsilon_s^2}$ to $\eta T$.

**Lemma C.3.** *Let $m$ be the total number of stages of Algorithm 1. We assume that $M \geq e$, where $M$ is a parameter of Algorithm 1. Then, we have*

$$\sum_{s=1}^{m} \frac{1}{\epsilon_s^2} \leq (\eta T)^2.$$

*Proof.* Essentially, this was proved in [Wan et al., 2023, Lemma 2] and easily follows from Lemma 2.2 and our problem setting. Since the number of queries of quantum reward oracles is limited up to $T$, by $C > 1$ and Lemma 2.2, we have

$$T \geq \sum_{s=1}^{m} \frac{\log(M/\delta)}{\eta \epsilon_s} \geq \eta^{-1} \sum_{s=1}^{m} \frac{1}{\epsilon_s} \geq \eta^{-1} \sqrt{\sum_{s=1}^{m} \frac{1}{\epsilon_s^2}}.$$

This completes the proof. $\qquad\square$

We can prove the following lemma by Corollary 5.6 and the proof of [Vakili et al., 2021, Corollary 1].

**Lemma C.4.** *Let $m$ be the total number of stages of Algorithm 1 and put $E = \sum_{s=1}^{m} \frac{1}{\epsilon_s^2}$.*

1. *Suppose that $k$ has $(C_p, \beta_p)$ polynomial eigendecay with $C_p > 0, \beta_p > 1$. Then, there exists a constant $c_p' > 0$ depending only on $C_p, \beta_p, \rho, \overline{k}, \overline{\psi}$ satisfying the following inequality:*

$$\gamma^{\mathrm{QMC}} \leq c_p' E^{1/\beta_p} \log^{1-1/\beta_p}(E + 1).$$

2. *Suppose that $k$ has $(C_{e,1}, C_{e,2}, \beta_e)$ exponential eigendecay with $C_{e,1}, C_{e,2}, \beta_e > 0$. Then, there exists a constant $c_e' > 0$ depending only on $C_{e,1}, C_{e,2}, \beta_e, \rho, \overline{k}, \overline{\psi}$ such that*

$$\gamma^{\mathrm{QMC}} \leq c_e' \log^{1+1/\beta_e}(E + 1).$$

*Proof.* First, suppose that $k$ has $(C_p, \beta_p)$ polynomial eigendecay. Then, by proof of [Vakili et al., 2021, Corollary 1], we have $\delta_D \leq C_p D^{1-\beta_p} \overline{\psi}^2$. By Corollary 5.6, we obtain

$$\gamma^{\mathrm{QMC}} \leq D \log\left(1 + \frac{\overline{k}}{D\rho} E\right) + \frac{C_p \overline{\psi}^2}{\rho} D^{1-\beta_p} E \leq D \log\left(1 + \frac{\overline{k}}{\rho} E\right) + \frac{C_p \overline{\psi}^2}{\rho} D^{1-\beta_p} E.$$

Taking $D = \lceil E^{1/\beta_p} \log^{-1/\beta_p}\left(1 + \frac{\overline{k}}{\rho} E\right) \rceil$, we see that $\gamma^{\mathrm{QMC}} \lesssim E^{1/\beta_p} \log^{1-1/\beta_p}(E + 1)$, where notation $\lesssim$ ignores constants depending only on $C_p, \beta_p, \rho, \overline{k}, \overline{\psi}$. Next, suppose that $k$ has $(C_{e,1}, C_{e,2}, \beta_e)$ exponential eigendecay. Then, by proof of [Vakili et al., 2021, Corollary 1], there exists a constant $c_e'' > 0$ depending only on $C_{e,1}, C_{e,2}, \beta_e, \overline{\psi}$ such that $\delta_D \leq c_e'' \exp\left(-C_{e,2}' D^{\beta_e}\right)$, where $C_{e,2}' = C_{e,2}$ if $\beta_e = 1$ and $C_{e,2}' = C_{e,2}/2$ if $\beta_e \neq 1$. Thus, by Corollary 5.6, we see that

$$\gamma^{\mathrm{QMC}} \leq D \log\left(1 + \frac{\overline{k}}{D\rho} E\right) + \frac{c_e''}{\rho} \exp\left(-C_{e,2}' D^{\beta_e}\right) E.$$

By taking $D = C_{e,2}'^{1/\beta_e} \lceil \log^{1/\beta_e}(E + 1) \rceil$, we have $\gamma^{\mathrm{QMC}} \lesssim \log^{1/\beta_e + 1}(E + 1)$, where notation $\lesssim$ hides constants depending only on $C_{e,1}, C_{e,2}, \beta_e, \rho, \overline{k}, \overline{\psi}$. $\qquad\square$

Similarly to [Wan et al., 2023], we relate $\gamma^{\mathrm{QMC}}$ to $m$.

**Lemma C.5.** *Let $m$ be the total number of stages of Algorithm 1 and $\gamma^{\mathrm{QMC}}$ be as in (3). Then, we have $\gamma^{\mathrm{QMC}} = m \log 2$.*

*Proof.* We let $\mathcal{G}_s$ be a finite dimensional subspace of $\mathcal{H}_k$ spanned by $\{\phi(x_1), \ldots, \phi(x_s)\}$ and $\mathcal{Q}_s = \sum_{i=1}^{s} \epsilon_i^{-2} \phi(x_i) \phi(x_i)^\top$. By Definition B.1 and its remark, we have

$$\gamma^{\mathrm{QMC}} = \log \det(I + \rho^{-1} \mathcal{Q}_m) = \log \det(I + \rho^{-1} \mathcal{Q}_m|_{\mathcal{G}_m}).$$

Therefore, the proof can be reduced to the finite dimensional case (i.e., the finite dimensional space $\mathcal{G}_m$) and by [Wan et al., 2023, Lemma 3], we have the following for each $s \geq 1$:

$$\det(I + \rho^{-1} \mathcal{Q}_s|_{\mathcal{G}_m}) = 2 \det(I + \rho^{-1} \mathcal{Q}_{s-1}|_{\mathcal{G}_m}).$$

Since $\mathcal{Q}_0 = 0$, we have the assertion of the lemma.

$\square$

*Proof of Proposition 5.3.* If $k$ has an exponential eigendecay and $\eta = 1$, then the statement of the proposition follows from Lemma C.3, C.4, C.5. Let us suppose $k$ has $\beta_p$ polynomial eigendecay. Then, by Lemma C.3, C.4, C.5, for any $\alpha > 0$, we have $m \lesssim (\eta T)^{2\beta_p^{-1}} \log^{1-\beta_p^{-1}} (1 + \eta T)$. By Proposition 5.2, we take $\eta$ so that $\eta^{-1}$ and $(\eta T)^{\beta_p^{-1}}$ have the same order, i.e., $\eta = T^{-1/(1+\beta_p)}$. This completes the proof.

$\square$

## C.6 PROOF OF THEOREM 5.4

*Proof.* Theorem 5.4 follows from Proposition 5.2 and Proposition 5.3.

$\square$

## D EXPERIMENTS

Proposition 5.3 and Theorem 5.4 imply that our regret bounds are better than that of [Dai et al., 2023] in the case of polynomial eigendecay due to the tradeoff parameter $\eta$. We note that Algorithm 1 with $\eta = 1$ is identical to Q-GP-UCB [Dai et al., 2023] and Proposition 5.3 suggests that the regret bound can be improved if we take the parameter $\eta$ as an appropriate small value. In this section, we empirically verify this in a simple synthetic environment using the quantum simulator provided by the qiskit library Javadi-Abhari et al. [2024].

In this environment, we define the quantum reward oracle $\mathcal{O}_x$ as a quantum circuit representing a Bernoulli random variable with mean $\mu(x) \in [0, 1]$, where the implementation is provided by the tutorial of qiskit-finace `https://qiskit-community.github.io/qiskit-finance/tutorials/00_amplitude_estimation.html`. For an implementation of QMC, we used the iterative amplitude estimation (IAE) [Grinko et al., 2021] implemented in the qiskit-algorithms library. Here, similar to [Dai et al., 2023], we used a theoretical upper bound of the number of oracle calls rather than the actual number of oracle calls of IAE. We consider a simple environment, where $T = 3000$, $\mathcal{X} = [0, 1]^d$ with $d = 1$, $k$ is the Matèrn-$\nu$ kernel with $\nu = 1.5$ and the length-scale 0.3, $\mu$ is given as $x \mapsto k(x, x_0)$ with $x_0 = 0.2$.

Since our algorithm in the case when $\eta = 1$ is identical to Q-GP-UCB [Dai et al., 2023], we have conducted an ablation study of the parameter $\eta$. Regarding the parameter of Algorithm 1, we take $\rho = 1.0, \delta/M = 10^{-2}, S = 1$. We run Algorithm 1 in the synthetic environment 10 times for each $\eta = 1.0, 10^{-1}, 10^{-2}, 10^{-3}$ and plot the cumulative regret in Figure 1, where the error bands represent 95% confidence intervals of cumulative regret. The experimental result supports our theoretical findings, i.e., by taking an appropriate (small) value of the parameter $\eta$, Algorithm 1 can achieve a better performance than the existing method [Dai et al., 2023]. Moreover, discussion in Section 4.3 suggests that by setting $\eta$ to a small value, the total number of stages decreases. In fact, in this experimental setting, the mean total number of stages when $\eta = 1$ is given as 200.8 (std 0.4) and that when $\eta = 10^{-2}$ is given as 18.6 (std 0.49).

For a better empirical performance, we introduce an exploration parameter $v > 0$ to the UCB

$$\widetilde{\mu}_{s-1}(x) + v\beta_{s-1}\widetilde{\sigma}_{s-1}(x). \tag{5}$$

We note that the case when $v = 1$ is identical to Algorithm 1. We conducted experiments using the UCB (5) with $v = 0.5, 0.1, 0.05$ with the same experimental setting and show cumulative regret in Figures 2 to 4. These experimental results also indicate that with an appropriate choice $\eta$, we have an improvement over Q-GP-UCB [Dai et al., 2023].

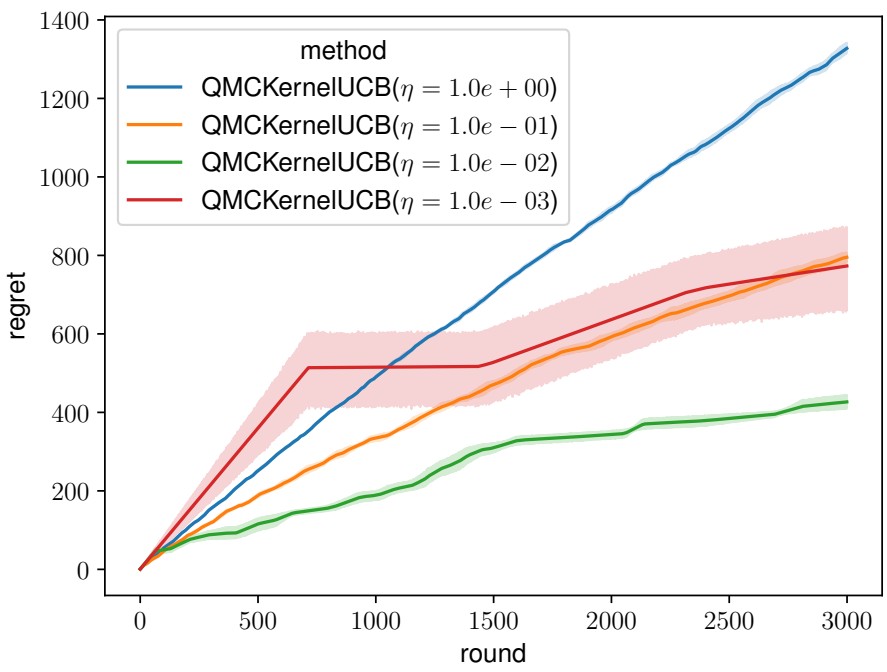

Figure 1: Ablation study of the parameter $\eta$. The case when $\eta = 1$ is identical to the existing method [Dai et al., 2023].

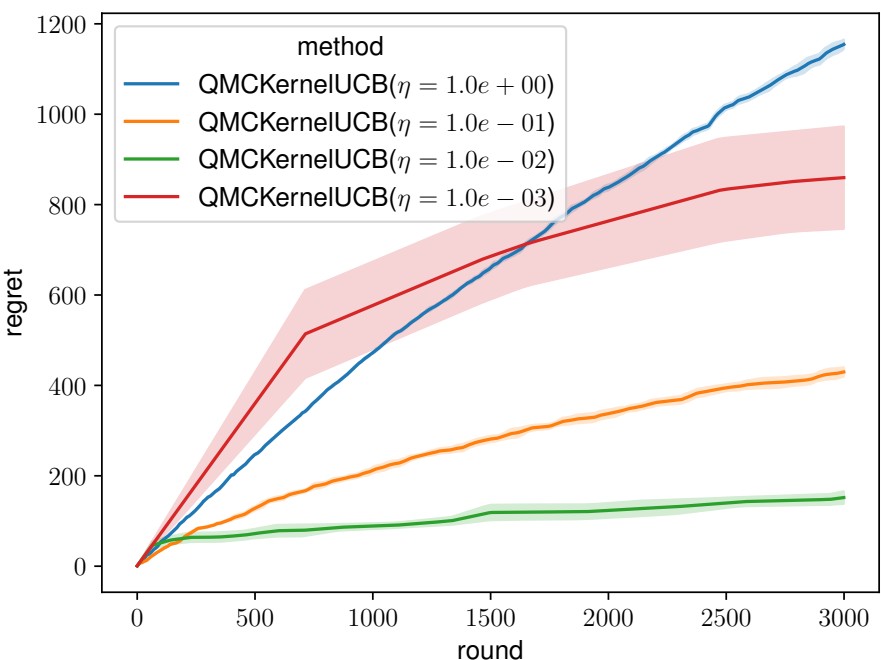

Figure 2: Ablation study of the parameter $\eta$ with the exploration parameter $v = 0.5$

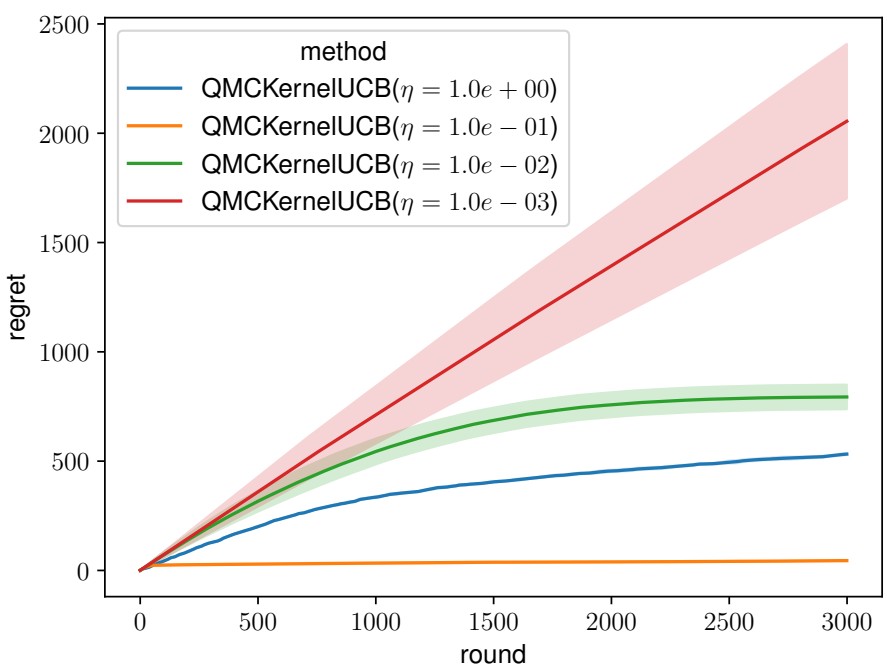

Figure 3: Ablation study of the parameter $\eta$ with the exploration parameter $v = 0.1$

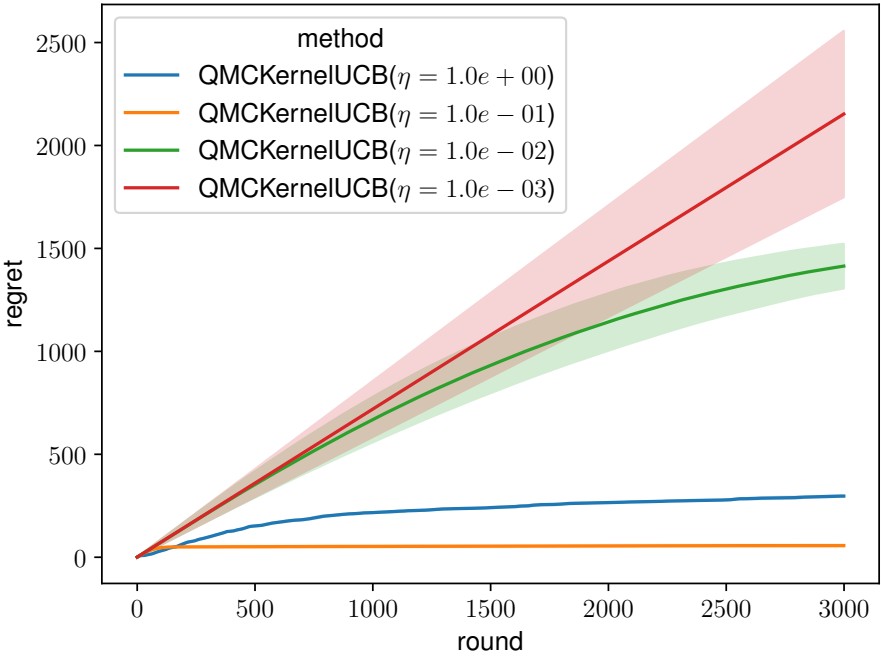

Figure 4: Ablation study of the parameter $\eta$ with the exploration parameter $v = 0.05$