# OpenReview forum: "Quantum Kernelized Bandits"
_auai.org/UAI/2024/Conference — UAI 2024 poster_

### Official Review · Reviewer_Nqga · 2024-02-28

**Q2-1 Originality-Novelty:** 3
**Q2-2 Correctness-Technical Quality:** 3
**Q2-5 Clarity Of Writing:** 3

**Q1 Summary And Contributions:**

This paper considers the quantum kernelized bandit problem, where the player observes information of a reward through quantum circuits called the quantum reward oracle, and the mean reward function belongs to an RKHS. The authors propose a UCB-type algorithm that utilizes the quantum Monte Carlo (QMC) method and develop regret bounds in terms of the decay rate of eigenvalues of the Mercer operator of the kernel. The authors establish high-probability regret bounds for cases when the kernel has a polynomial eigendecay and an exponential eigendecay, respectively. In particular, in the case of the exponential eigendecay, the provided regret bound exponentially improve that of classical algorithms. Moreover, the provided regret bound is better than the lower bound in the classical kernelized bandit problem if the rate of decay is sufficiently fast.

**Q2-3 Extent To Which Claims Are Supported By Evidence:**

3: Good: the main claims are supported by convincing evidence (in the form of adequate experimental evaluation, proofs, (pseudo-)code, references, assumptions).

**Q2-4 Reproducibility:**

3: Good: key resources (e.g. proofs, code, data) are available and key details (e.g. proofs, experimental setup) are sufficiently well-described for competent researchers to confidently reproduce the main results.

**Q3 Main Strengths:**

1.	The studied problem, quantum kernelized bandits, is very interesting to the bandit and quantum machine learning literatures, and can inspire potential future works.
2.	The authors propose a UCB-type algorithm that utilizes the quantum Monte Carlo method with rigorous theoretical analysis.
3.	The authors derive regret bounds in terms of the decay rate of eigenvalues of the Mercer operator of the kernel for two cases, i.e., a polynomial eigendecay and an exponential eigendecay. The presented regret bounds are better than that for classic kernelized bandit algorithms, and even the lower bound for the classical kernelized bandit problem if the rate of decay is sufficiently fast.

**Q4 Main Weakness:**

The authors should give a more detailed discussion on their technical novelty compared to existing quantum bandit works, apart from that this work considers the kernelized setting.

In other words, it is important to justify why the contribution is not a straightforward combination of the classic kernelized bandit results and the quantum online learning techniques.

**Q5 Detailed Comments To The Authors:**

Please see the weaknesses above.

**Q9 Complying With Reviewing Instructions:**

Yes

---

> ### Author Rebuttal · Authors · 2024-04-05
>
> We sincerely thank the reviewer for taking time and effort in providing valuable feedback on our manuscript. We will improve our manuscript based on your suggestion.
>
> ## Comparison to existing quantum online learning algorithms
>
> The most relevant papers are Wan et al. [2023] (the linear case) and Dai et al. [2023] (the kernelized case under the unbiasedness assumption of the QMC estimator), and we believe that we have provided sufficient comparison in the submitted paper.
> Other than that, for instance, Li and Zhan [2022] studied the quantum bandit convex optimization and Wang et al. [2021] studied a best arm identification problem in the quantum multi-arm bandit setting. Although these algorithms are also stage-based algorithms, they are quite different from ours due to the different problem settings (for example, Li and Zhan [2022] proposed a quantum analogue of a simulated annealing algorithm). Therefore, their theoretical analysis are quite different from ours and we cannot simply combine these existing quantum online algorithms with the classical kernelized bandits. We appreciate your valuable feedback and in the final version of the paper, we will detail this in the related work section.

---

### Official Review · Reviewer_sD9Z · 2024-03-21

**Q2-1 Originality-Novelty:** 3
**Q2-2 Correctness-Technical Quality:** 3
**Q2-5 Clarity Of Writing:** 2

**Q1 Summary And Contributions:**

This paper consider quantum kernelized bandit problem and propose QMCkernelUCB. The authors also provide regret analysis for the proposed algorithm.

**Q2-3 Extent To Which Claims Are Supported By Evidence:**

2: Fair: the main claims are somewhat supported by evidence (but the experimental evaluation may be weak, or does not match entirely with the claims, important baselines may be missing, proofs contain important ideas but lack rigor, algorithmic details are only discussed superficially, references are imprecise, assumptions are not sufficiently motivated or explicated, etc.).

**Q2-4 Reproducibility:**

2: Fair: key resources (e.g. proofs, code, data) are unavailable but key details (e.g. proof sketches, experimental setup) are sufficiently well-described for an expert to confidently reproduce the main results.

**Q3 Main Strengths:**

1. Novel bandit problem with a practical proposed algorithm.

2. Comprehensive regret analysis.

**Q4 Main Weakness:**

1. No empirical study.

2. Presentation is not easy to follow.

**Q5 Detailed Comments To The Authors:**

Just curious about the following points:

1. what exactly is QMC? Hot to perform line 4 in Algorithm 1?

2. What's the role of line5-7 in algorithm 1? It looks like taking the same action repeatedly.

3. Optimization in Eq(2) is not a ridge loss but a weighted MSE with $\ell_2$ penalty. Can you explain why it has to be the weights $1/\epsilon_i^2$? How to compute $\epsilon_i^2$?

**Q9 Complying With Reviewing Instructions:**

Yes

---

> ### Author Rebuttal · Authors · 2024-04-05
>
> We sincerely thank the reviewer for taking time and effort in providing valuable feedback on our manuscript. We will improve our manuscript based on your suggestion.
>
> ## Empirical study
> We have conducted an experiment in a synthetic environment on a quantum simulator. Since comparisons with classical bandit algorithms have been conducted by the previous work [Dai et al. 2023], we compare our method to Dai et al. [2023]. In the following we show cumulative regret of our method QMCKernelUCB for several choices of the parameter $\eta$ in a synthetic environment ($\mathcal{X} = [0, 1]^d, d=1$, $k$: Matern kernel with $\nu=1.5$, $\rho=10^{-2}$).
> We note that the case when $\eta=1$ is identical to the method of Dai et al. [2023]. Our theoretical result (Prop. 5.5) indicate we should set $\eta$ to an appropriate small value and the experimental results show that the performance with $\eta=10^{-2}$ is better than Dai et al. [2023]. We will add more details of the experiments and figures in the final version of the paper.
>
> | $\eta$ | regret (t=200) | regret (t=400) |  regret (t=600)|regret (t=800)|
> |----|----|----|----|----|
> |1.0e+00 (Dai et al. [2023]) | 1e+02 ($\pm$ 2e+00)|2e+02 ($\pm$ 2e+00)|3e+02 ($\pm$ 2e+00)|4e+02 ($\pm$ 4e+00)|
> |1.0e-01 (Ours) | 9e+01 ($\pm$ 4e+00)|2e+02 ($\pm$ 5e+00)|2e+02 ($\pm$ 1e+01)|3e+02 ($\pm$ 1e+01)|
> |1.0e-02 (Ours) | 6e+01 ($\pm$ 2e+01)|8e+01 ($\pm$ 2e+01)|1e+02 ($\pm$ 9e+00)|1e+02 ($\pm$ 2e+01)|
> |1.0e-03 (Ours) | 1e+02 ($\pm$ 4e+01)|3e+02 ($\pm$ 8e+01)|4e+02 ($\pm$ 1e+02)|6e+02 ($\pm$ 2e+02)|
>
> ## Presentation of the paper
> We will add more detailed, intuitive explanation of our method in the final version of the paper.
>
> ## Response to the questions in Q5
> > what exactly is QMC? Hot to perform line 4 in Algorithm 1?
>
> QMC is an algorithm that satisfies the properties of Lemma 2.2 and its pseudo code (or that of an improved algorithm) is given by Montanaro [2015] and subsequent papers.
> One can implement QMC using the quantum amplitude estimation (QAE) algorithm (c.f., Rebentrost et al. [2018]). Several implementations (such as AmplitudeEstimation, IterativeAmplitudeEstimation) of QAE is provided by the qiskit library, which is a Python library developed by IBM. To perform line 4, one can utilize these implementations. However, at the moment, one can perform QMC only on a quantum simulator since QMC (or QAE) requires a (large) fault tolerant quantum computer.
>
> > What's the role of line5-7 in algorithm 1? It looks like taking the same action repeatedly.
>
> Yes, it selects the same action repeatedly during successive $n\_s := \frac{C}{\eta\epsilon\_s}\log M/\delta$ rounds in stage $s$. It is important to note that in the setting of the quantum bandit problem, by taking an action $x\_t$, the learner only incurs instantaneous regret $\mu(x^*) - \mu(x\_t)$ and unlike the classical bandit problem, the leaner may not observe information about reward $\mu(x\_t)$ in every round $t$. For instance, by calling the QMC method, the leaner takes the same action during successive $n\_s$ rounds, and observes only one output $y\_s$ (we note that $n\_s$ is equal to the upper bound of the number of oracle queries of the QMC method). By performing a measurement in every round, the learner can observe information about reward in every round, however, an estimation of QMC is more sample efficient than this naive method.
>
> > Optimization in Eq(2) is not a ridge loss but a weighted MSE with $\ell^2$ penalty. Can you explain why it has to be the weights $1/\epsilon_i^2$? How to compute $\epsilon_i^2$?
>
> Since $y\_i$ is an estimation of $\mu(x\_i)$ returned by the QMC method with an error tolerance $\eta \epsilon\_i$, the estimation $y\_i$ is more reliable if $\epsilon\_i$ is the smaller. Therefore, we consider the weighted least squares estimator with weights $\epsilon\_i^{-2}$. We can compute $\epsilon\_i^2$ by the kernel trick (Proposition 4.1). In the final version of the paper, we will detail an intuitive explanation of our method in Section 4.

---

### Official Review · Reviewer_77Vf · 2024-03-22

**Q2-1 Originality-Novelty:** 2
**Q2-2 Correctness-Technical Quality:** 3
**Q2-5 Clarity Of Writing:** 3

**Q1 Summary And Contributions:**

This paper considers a bandit problem setting in which the arms in a bandit instance can be queried via quantum oracles, and the mean reward function belongs to an RKHS. The paper makes several novel contributions compared with the recent work of quantum Bayesian optimization, including getting rid of the requirement for the additional assumption of the unbiasedness of the quantum MC estimator.

**Q2-3 Extent To Which Claims Are Supported By Evidence:**

2: Fair: the main claims are somewhat supported by evidence (but the experimental evaluation may be weak, or does not match entirely with the claims, important baselines may be missing, proofs contain important ideas but lack rigor, algorithmic details are only discussed superficially, references are imprecise, assumptions are not sufficiently motivated or explicated, etc.).

**Q2-4 Reproducibility:**

3: Good: key resources (e.g. proofs, code, data) are available and key details (e.g. proofs, experimental setup) are sufficiently well-described for competent researchers to confidently reproduce the main results.

**Q3 Main Strengths:**

The paper provides a more general proof technique for the confidence interval in quantum kernelized bandits, which compared to the previous work of Dai et al. (2023), does not require the additional assumption of the quantum Monte Carlo algorithm being unbiased. The introduction of the additional tuning parameter $\eta$ provides the flexibility to trade-off the regrets in every stage and the total number of stages, and therefore also helps derive a more general upper bound on the regret,.

**Q4 Main Weakness:**

- One of the most novel aspect (perhaps the most novel aspect) of the paper is an alternative proof of the confidence interval without requiring the assumption of an unbiased QMC estimator. However, in my opinion, it may not be unreasonable to assume that the adopted QMC algorithm is unbiased. Since the development of the original quantum amplitude estimation algorithm, there have been improved versions for QMC which can make the QMC estimation unbiased. One such example is the paper "On the bias in iterative quantum amplitude estimation, 2023", as well as the relevant papers referenced therein. This may put the novelty of the contributions of this paper into question.
- I think it would be good to add some discussions on the comparisons of the theoretical results with those from Dai et al. (2023), mostly comparisons of the confidence intervals and the final upper bounds on the regret. I understand that since Dai et al. (2023) requires an additional assumption, so that their results may be better, but it would be interesting to see how much can be gained from this assumption.
- If I understand correctly, Algorithm 1 in this paper is the same as the Q-GP-UCB algorithm from Dai et al. (2023), except for the parameter $\eta$? I wonder is there any other novelty in terms of the design of the algorithm.
- From what I have seen, the previous paper on quantum bandits often use some simulations to test the developed algorithm. So I think it would be good to add some experiments. For example, an ablation study to test the impact of $\eta$ would be good.

**Q5 Detailed Comments To The Authors:**

- At the beginning of Section 3.2, it is assumed that the value of the mean reward function is bounded within [0,1]. Is this a common assumption in kernelized bandits? Similarly, in Assumption 5.3 (d), I understand that it is common to have such an assumption in linear bandits, but is it also a common assumption in kernelized bandits?
- Regarding the weighted information gain, the paper mentions that a more generalized proof technique is used in this paper. I wonder are the results also more general than those from Vakili et al. (2021) and Dai et al. (2023)?

**Q9 Complying With Reviewing Instructions:**

Yes

---

> ### Author Rebuttal · Authors · 2024-04-05
>
> We sincerely thank the reviewer for taking time and effort in providing detailed, insightful feedback on our manuscript. We will improve our manuscript based on your suggestion and this discussion.
>
> ## Bias of the QMC estimator
> Regarding recent improved QAE (quantum amplitude estimation) methods, in our understanding, Miyamoto [2023] and related work only proposed bias-mitigation methods for QAE (or IQAE) and they do not claim that the estimator is unbiased. Even if the estimator is approximately unbiased, these improved methods require a larger number of queries (in the experiment in Miyamoto [2023], the increase rate is around 20%). Therefore, our paper provides theoretical guarantee for a wide class of QMC implementations.
>
> **References**
>
> Myamoto [2023], On the bias in iterative quantum amplitude estimation, 2023
> ## Theoretical comparison to Dai et al. [2023]
> Our response is two-fold.
> First, by assuming the unbiasedness of the QMC estimator, we can show our method is better than the regret bound of Dai et al. [2023] in the case of Matern kernels (kernels with polynomial eigendecay) due to the novel feature of our algorithm.
> Second, surprisingly, we found that the aforementioned improvement can be obtained without assuming the unbiasedness of the estimator.
>
> We explain more details as follows. First, by assuming the unbiasedness of the QMC estimator, the confidence interval (Prop. 4.2) can be improved to $|\mu(x) - \tilde{\mu}\_s(x)| \lesssim (1 + \eta \sqrt{m})\tilde{\sigma}\_s(x)$.
> Therefore, Prop. 5.4 can be improved to $R(T) = \widetilde{O}(m (\eta^{-1} + \sqrt{m}))$.
> Thus, by setting $\eta = \Theta(m^{-1/2})$, in the case of $\beta_p$-polynomial eigendecay, the regret bound can be improved to $R(T) = \widetilde{O}(T^{3/(1 + \beta\_p)})$, which improves the regret bound $\widetilde{O}(T^{3/\beta\_p})$ of Dai et al. [2023] due to the novel tradeoff parameter.
> We also note that our regret bound $R(T) = \widetilde{O}(T^{3/(1 + \beta\_p)})$ is better than regret bound $\widetilde{O}(T^{1/2 + 1/\beta\_p})$ of GP-UCB whenever GP-UCB has sublinear regret (i.e., $\beta\_p > 2$), while that of  Dai et al. [2023] is not necessarily better than GP-UCB.
>
> Second, after our submission, surprisingly, we found that we can prove the confidence interval of the form $|\mu(x) - \tilde{\mu}\_s(x)| \lesssim (1 + \eta \sqrt{m})\tilde{\sigma}\_s(x)$ even if we do not assume the unbiasedness of the estimator. Although the proof technique is standard, by combining with our novel algorithm, we obtain the aforementioned improvement over Dai et al. [2023]. We provide a sketch of the proof in the following.
>
> #### Sketch of the proof of the improved confidence interval
> We use notation of Proof of Lemma C.2. To prove the improved confidence interval, it is sufficient to prove $\|\Phi\_s^\top W\_s^{1/2} \Gamma\_s\|\_{V\_{s}^{-1}}^2 \le \eta^2 s$.
> By the same proof as in Lemma C.2, we have
> $\|\Phi\_s^\top W_s^{1/2} \Gamma_s\|\_{V\_{s}^{-1}}^2 \le \mathrm{Tr} \left(\Gamma\_s \Gamma\_s^\top\right)\sigma\_{\mathrm{max}} \left(  W\_s^{1/2} \Phi\_s V\_s^{-1} \Phi\_s^\top W\_s^{1/2}\right)$, where $\mathrm{Tr} \left(\Gamma\_s \Gamma\_s^\top\right) \le \eta^2s$.
> By the standard argument (c.f. Valko et al. [2013]), one can prove
> $\sigma\_{max} \left( W\_s^{1/2} \Phi\_s V\_s^{-1} \Phi\_s^\top W\_s^{1/2}\right)\le 1$.
> ## Novelty of Algorithm 1
> Yes, on the algorithmic side, the trade-off parameter is the only difference compared to Dai et al. [2023]. However, this key feature provides the aforementioned improvement.
>
> ## Experimental Results
> We have conducted an experiment in a synthetic environment ($\mathcal{X} = [0, 1]^d, d=1$, $k$: Matern kernel with $\nu=1.5$, $\rho=10^{-2}$) using the qiskit library and `IterativeAmplitudeEstimation`.
> The case when $\eta=1$ corresponds to the method of Dai et al. [2023] and we show cumulative regret at $t=200, 400, 600, 800$ for $\eta=1,10^{-2}$. Our theoretical result (Prop. 5.5) indicates we should set $\eta$ to an appropriate small value and the experimental results shows that the performance with $\eta=10^{-2}$ is better than that with $\eta=1$. We refer to the response to other reviewers for an extended table.
>
> | $\eta$ | regret (t=200) | regret (t=400) |  regret (t=600)|regret (t=800)|
> |----|----|----|----|----|
> |1.0 | 1e+02 ($\pm$ 2e+00)|2e+02 ($\pm$ 2e+00)|3e+02 ($\pm$ 2e+00)|4e+02 ($\pm$ 4e+00)|
> | $10^{-2}$ | 6e+01 ($\pm$ 2e+01)|8e+01 ($\pm$ 2e+01)|1e+02 ($\pm$ 9e+00)|1e+02 ($\pm$ 2e+01)|
>
> ## Bounded reward and generalized proof technique
> The standard boundedness assumption [Chowdhury and Gopalan, 2017] implies rewards are bounded by Cauchy-Schwarz. Therefore, after normalization (affine transformation) of rewards, the standard assumption implies our assumption. In Sec. 5.3, we show how we obtain the result of Dai et al. [2023] as a corollary of Prop. 5.7. The case of Vakili et al. [2021] is the same. We will add more details in the final version.

---

### Official Review · Reviewer_8Lah · 2024-03-23

**Q2-1 Originality-Novelty:** 3
**Q2-2 Correctness-Technical Quality:** 3
**Q2-5 Clarity Of Writing:** 3

**Q1 Summary And Contributions:**

The paper considers the quantum kernelized bandit problem where the players obtained information about a reward through a quantum reward oracle, and the mean reward function belongs to a reproducing kernel Hilbert space. They proposed a UCB-type algorithm for quantum bandit problems where the reward function is non-linear with respect to an action. By employing Mercer’s theorem, they provided a theoretical analysis that the proposed algorithm achieves O(poly(log T)) regret bound when the decay rate of the Mercer operator decreases exponentially fast.

**Q2-3 Extent To Which Claims Are Supported By Evidence:**

3: Good: the main claims are supported by convincing evidence (in the form of adequate experimental evaluation, proofs, (pseudo-)code, references, assumptions).

**Q2-4 Reproducibility:**

3: Good: key resources (e.g. proofs, code, data) are available and key details (e.g. proofs, experimental setup) are sufficiently well-described for competent researchers to confidently reproduce the main results.

**Q3 Main Strengths:**

This paper extends the quantum linear bandit problem to the kernelized case. They show the case where the rate of decay of eigenvalues of the Mercer operator is polynomially or exponentially fast and provides an upper bound of the cumulative regret. They show that the proposed method exponentially improves over the classical algorithms under the condition of the exponential eigendecay.

The paper is theoretical in nature and well-written. Most of the proofs are deferred to the appendix. The proof sketch looks believable. However, I didn't get time to verify them.

**Q4 Main Weakness:**

I have some minor comments on the writing style, though it's optional for the authors.

1) Section 5.1 is based on known facts - so it may be moved to Section 2.

2) COMPARISON TO Dai et al. [2023] can be combined with related work.

**Q5 Detailed Comments To The Authors:**

Pls see in response to Q4.

**Q9 Complying With Reviewing Instructions:**

Yes

---

> ### Author Rebuttal · Authors · 2024-04-05
>
> We sincerely thank the reviewer for taking time and effort in providing valuable feedback on our manuscript. In the final version of the paper, we will move Section 5.1 to Section 2 or 3, and "COMPARISON TO Dai et al. [2023]" (Section 4.3) to the related work section.

---

### Official Review · Reviewer_mnpf · 2024-03-23

**Q2-1 Originality-Novelty:** 3
**Q2-2 Correctness-Technical Quality:** 3
**Q2-5 Clarity Of Writing:** 4

**Q1 Summary And Contributions:**

This paper studies the kernelized linear bandits with quantum computing. They propose a LIN-UCB based algorithm, QMC-Kernel-UCB and achieve a better regret bound as compared to the counterpart without using quantum computing. The improvement from the fact that quantum concentration has a faster rate.

**Q2-3 Extent To Which Claims Are Supported By Evidence:**

4: Excellent: all claims are supported by very convincing evidence (in the form of comprehensive experimental evaluation, rigorous mathematical proofs, detailed (pseudo-)code, precise references, well-motivated and realistic assumptions) and the authors deliver what they promise.

**Q2-4 Reproducibility:**

4: Excellent: key resources (e.g. proofs, code, data) are available and key details (e.g. proof sketches, experimental setup) are comprehensively described for competent researchers to confidently and easily reproduce the main results.

**Q3 Main Strengths:**

1. They identify some proofs error in previous work and present correct proof. This is important to all future work related to quantum bandit.

2. Algorithm their proposed algorithm extend LIN-UCB to quantum concentration, the proof is non-trivial.

**Q4 Main Weakness:**

1. Their proposed algorithm still needs to do matrix inversion, which is still computationally expensive. It is worthwhile to design algorithms without matrix inversion, for example, LMC-TS based (https://arxiv.org/abs/2206.11254).

2. It will be good to have some experiments to see the performance empirically.

3. Since quantum computing is to speed up learning tasks, it will be good if it can be used in deep neural networks.

**Q5 Detailed Comments To The Authors:**

n/a

**Q9 Complying With Reviewing Instructions:**

Yes

---

> ### Author Rebuttal · Authors · 2024-04-05
>
> We sincerely thank the reviewer for taking time and effort in providing insightful suggestions on our manuscript. We will improve our manuscript based on your suggestion.
>
> ## Algorithms without matrix inversion
> Regarding the algorithm design, it would be an intriguing extension to explore the possibility of designing an algorithm that does not necessitate the computation of matrix inversion, such as LMC-TS which is based on noisy gradient descent. We are keen on considering more efficient algorithms in our future research and will discuss this in the future work section.
>
> ## Experimental results
> We agree with the significance of experimentally validating the performance of the algorithms proposed in the present study. Specifically, it would be beneficial to examine how the parameter $\eta$ of the algorithm, which is the focal point of this study, affects the regret results. Experimental results under a synthetic environment are reported below.
>
> In the following we show cumulative regret of our method QMCKernelUCB for several choices of $\eta$ in a synthetic environment ($\mathcal{X} = [0, 1]^d, d=1$, $k$: Matern kernel with $\nu=1.5$, $\rho=10^{-2}$) using the qiskit library and `IterativeAmplitudeEstimation`. Since comparisons with classical bandit algorithms have been conducted by the previous work [Dai et al. 2023], we compare our method to Dai et al. [2023]. We note that the case when $\eta=1$ is identical to the method of Dai et al. [2023]. Our theoretical result (Prop. 5.5) indicates we should set $\eta$ to an appropriate small value and the experimental results shows that the performance with $\eta=10^{-2}$ is better than Dai et al. [2023]. We will add more details of the experiments add figures in the final version of the paper.
>
> | $\eta$ | regret (t=200) | regret (t=400) |  regret (t=600)|regret (t=800)|
> |----|----|----|----|----|
> |1.0e+00 (Dai et al. [2023]) | 1e+02 ($\pm$ 2e+00)|2e+02 ($\pm$ 2e+00)|3e+02 ($\pm$ 2e+00)|4e+02 ($\pm$ 4e+00)|
> |1.0e-01 (Ours) | 9e+01 ($\pm$ 4e+00)|2e+02 ($\pm$ 5e+00)|2e+02 ($\pm$ 1e+01)|3e+02 ($\pm$ 1e+01)|
> |1.0e-02 (Ours) | 6e+01 ($\pm$ 2e+01)|8e+01 ($\pm$ 2e+01)|1e+02 ($\pm$ 9e+00)|1e+02 ($\pm$ 2e+01)|
> |1.0e-03 (Ours) | 1e+02 ($\pm$ 4e+01)|3e+02 ($\pm$ 8e+01)|4e+02 ($\pm$ 1e+02)|6e+02 ($\pm$ 2e+02)|
>
> ## Deep neural networks
> We acknowledge your view that quantum computing is anticipated to accelerate deep neural networks as well as machine learning algorithms.
> So far, many quantum machine learning methods such as quantum support vector machines and quantum principal component analysis have been developed and demonstrated superior efficiency over classical counterparts. In recent years, research has also been explored to construct classical deep neural networks on quantum computers, and it has been reported quantum acceleration under certain conditions (e.g., [1]). Moreover, RKHS (or kernel methods) is closely linked to deep neural networks via neural tangent kernel and the kernelized bandits are generalized to online learning of deep neural networks [2]. Thus, as the reviewer commented, quantum technology could potentially contribute to the speed-up of learning tasks related to deep neural networks.
>
>
> **References**
>
> [1] Zhao, Chen, and Xiao-Shan Gao. "QDNN: DNN with quantum neural network layers." arXiv preprint arXiv:1912.12660 (2019).
>
> [2] Zhou, Dongruo, Lihong Li, and Quanquan Gu. "Neural contextual bandits with ucb-based exploration." International Conference on Machine Learning. PMLR, 2020.

---

### Meta-Review · Area_Chair_ViLr · 2024-04-15

The reviewers agreed that the contributions of this paper are significant compared with existing results.